# EquiformerV3: Scaling Efficient, Expressive and General SE(3)-Equivariant Graph Attention Transformers

## Abstract

As $SE(3)$-equivariant graph neural networks mature as a core tool for 3D atomistic modeling, improving their efficiency, expressivity, and physical consistency has become a central challenge for large-scale applications. In this work, we introduce EquiformerV3, the third generation of the $SE(3)$-equivariant graph attention Transformer, designed to advance all three dimensions: efficiency, expressivity, and generality. Building on EquiformerV2, we have the following three key advances. First, we optimize the software implementation, achieving $1.75\times$ speedup. Second, we introduce simple and effective modifications to EquiformerV2, including equivariant merged layer normalization, improved feedforward network hyper-parameters, and attention with smooth radius cutoff. Third, we propose SwiGLU-$S^2$ activations to incorporate many-body interactions for better theoretical expressivity and to preserve strict equivariance while reducing the complexity of sampling $S^2$ grids. Together, SwiGLU-$S^2$ activations and smooth-cutoff attention enable accurate modeling of smoothly varying potential energy surfaces (PES), generalizing EquiformerV3 to tasks requiring energy-conserving simulations and higher-order derivatives of PES. With these improvements, EquiformerV3 achieves state-of-the-art results on OC20, OMat24, and Matbench Discovery.

## 1. Introduction

Machine learning (ML) has enabled major advances in accelerating and scaling highly accurate but computationally intensive quantum mechanical calculations, such as density functional theory (DFT), for 3D atomistic systems (Gilmer et al., 2017; Zhang et al., 2018; Unke et al., 2021; Batzner et al., 2022; Rackers et al., 2023; Lan et al., 2022), reducing evaluation times from hours or days to fractions of a second. This capability opens the door to fast and accurate simulation pipelines that can drive progress in molecular dynamics (Musaelian et al., 2023), catalyst design (Lan et al., 2022) and materials discovery (Riebesell et al., 2024). Meanwhile, the rapid growth of open 3D atomistic datasets and community benchmarks, including QM9 (Ramakrishnan et al., 2014), MD17 (Chmiela et al., 2017), OC20 (Chanussot* et al., 2021), OC22 (Tran* et al., 2022), ODAC23 (Sriram et al., 2024), OMat24 (Barroso-Luque et al., 2024), Matbench Discovery (Riebesell et al., 2024), and OMol25 (Levine et al., 2025), has enabled systematic evaluation of modeling approaches at scale. Across these benchmarks, $SE(3)$-equivariant graph neural networks (GNNs) (Thomas et al., 2018; Kondor et al., 2018; Fuchs et al., 2020; Batzner et al., 2022; Brandstetter et al., 2022; Musaelian et al., 2022; Liao & Smidt, 2023; Passaro & Zitnick, 2023) have emerged as a particularly effective and reliable class of models. $SE(3)$-equivariant GNNs represent atomistic systems as graphs, naturally capturing the set-like structure of atomic configurations. They incorporate inductive biases that enforce equivariance of internal features and outputs under 3D rotations and translations. In practice, these models operate on vector spaces of irreducible representations (irreps), using equivariant operations such as tensor products to construct message passing, aggregation, and nonlinear transformations that respect the underlying symmetries of the physical system.

Progress in equivariant GNNs has been driven by advances along several complementary directions. These include the design of more expressive nonlinearities, such as gate activations (Weiler et al., 2018) and $S^2$ activations (Cohen et al., 2018); improved normalization schemes, including equivariant layer normalization (Liao & Smidt, 2023) and separable layer normalization (Liao et al., 2024b); and mechanisms for incorporating many-body interactions through self tensor products (Batatia et al., 2022). Substantial efforts have also focused on reducing the computational cost of tensor product operations, through methods such as fast spherical harmonic products (Xin et al., 2021), eSCN (Passaro & Zitnick, 2023), Gaunt tensor products (Luo et al., 2024),

---

[1]Anonymous Institution, Anonymous City, Anonymous Region, Anonymous Country. Correspondence to: Anonymous Author <anon.email@domain.com>.

Preliminary work. Under review by the International Conference on Machine Learning (ICML). Do not distribute.

and the Price of Freedom framework (Xie et al., 2025). In parallel, increasingly effective architectural designs have emerged that combine these components into powerful models (Liao & Smidt, 2023). Among these, Equiformer (Liao & Smidt, 2023) provides a framework that can integrate most advances in equivariant neural networks with Transformer architectures that have proven successful in natural language processing (Vaswani et al., 2017) and computer vision (Dosovitskiy et al., 2021). Building on this foundation, subsequent models such as EquiformerV2 (Liao et al., 2024b), eSEN (Fu et al., 2025), and UMA (Wood et al., 2025) have consistently advanced the state of the art across major atomistic benchmarks and leaderboards.

In this paper, we present EquiformerV3[1], the **third** generation of **equi**variant graph attention Trans**former** (or Equiformer), which further pushes the efficiency, expressivity, and generality of equivariant GNNs. Building on EquiformerV2, we introduce three key improvements. First, we optimize the software implementation of EquiformerV2 by fusing redundant operations and enabling compilation, resulting in $1.75\times$ speedup. Second, we adopt simple and effective modifications to operations in EquiformerV2 for better performance and generality. These include equivariant merged layer normalization, better architectural hyperparameters for feedforward networks, and attention with smooth radius cutoff. Third, we propose SwiGLU-$S^2$ activation function, which incorporates fast tensor products on $S^2$ grids into the activation function. This improvement enables many-body interactions for greater theoretical expressivity and preserves strict equivariance while reducing the complexity of sampling $S^2$ grids. Particularly, combining the SwiGLU-$S^2$ activation and the smooth radius cutoff in attention enables modeling smoothly varying potential energy surface (PES), generalizing EquiformerV3 to tasks requiring energy-conserving simulations and higher-order derivatives of PES.

With these improvements, EquiformerV3 demonstrates better efficiency and stronger performance than EquiformerV2 (Liao et al., 2024b) and sets state-of-the-art results on OC20 (Chanussot* et al., 2021), OMat24 (Barroso-Luque et al., 2024), and Matbench Discovery (Riebesell et al., 2024). On the OC20 S2EF-2M dataset, EquiformerV3 demonstrates up to $5.9\times$ speedup in training efficiency. On OMat24, EquiformerV3 with maximum degree $L_{max} = 4$ can achieve comparable force MAE to EquiformerV2 and UMA-L (Wood et al., 2025) while being $5\times$ and $23\times$ smaller in terms of model sizes, respectively. On Matbench Discovery, EquiformerV3 demonstrates $18\%$ to $31\%$ improvements in the thermal conductivity task compared to eSEN (Fu et al., 2025).

---

[1]We encourage using EquiformerV3 to represent this work instead of other terms like EqV3 and eqV3.

Moreover, EquiformerV3 simultaneously obtains the best results on all the metrics considered in the combined performance score (CPS) and saves $22.6\times$ training time compared to UMA-M.

## 2. Background

We briefly discuss the relevant background on $SE(3)$-equivariant networks and the Equiformer series here. Additional related works are discussed in Section A.

### 2.1. $SE(3)$-Equivariant Neural Networks

Including equivariance in neural networks as an inductive bias can improve data efficiency and generalization. For 3D atomistic systems, the relevant symmetries are 3D rotation, translation and inversion in 3D space. These transformations form the Euclidean group $E(3)$, with rotations and translations alone forming the special Euclidean group $SE(3)$. Since all the operations in EquiformerV3 are $SE(3)$-equivariant, we focus on $SE(3)$-equivariance and do not explicitly consider inversion symmetry.

Translation equivariance (or, more precisely, invariance) is achieved by operating on relative positions between atoms. Rotational equivariance is enforced by representing features in vector spaces of irreducible representations (irreps) of $SO(3)$. Each irrep is characterized by a non-negative integer degree $L$ and has dimension $(2L + 1)$. Intuitively, $L$ can be interpreted as an angular frequency, determining how rapidly the representation transforms under rotations of the coordinate system. Higher values of $L$ encode finer angular structure and are essential for tasks that depend sensitively on directional information, such as force prediction (Batzner et al., 2022; Zitnick et al., 2022; Passaro & Zitnick, 2023). We refer to vectors transforming according to the degree-$L$ irrep as type-$L$ vectors. Under a rotation, they transform via the Wigner-$D$ matrices $D^{(L)}$. Euclidean vectors $\vec{r} \in \mathbb{R}^3$ can be projected into type-$L$ vectors using spherical harmonics $Y^{(L)}\big(\frac{\vec{r}}{||\vec{r}||}\big)$. Each type-$L$ vector has components indexed by the order $m$, where $-L \leq m \leq L$. An equivariant irreps feature $f$ is constructed by concatenating multiple type-$L$ vectors across different degrees. Specifically, $f$ contains $C_L$ channels of type-$L$ vectors for $0 \leq L \leq L_{max}$, where $C_L$ denotes the number of channels for degree $L$. In this work, we use a uniform channel size $C_L = C$ for all degrees, so the total feature dimension is $(L_{max} + 1)^2 \times C$. We index $f$ by channel $i$, degree $L$, and order $m$, and denote its components as $f_{m,i}^{(L)}$.

Equivariant networks update irreps features using equivariant operations. In practice, interactions between different type-$L$ features are realized through tensor products. We denote a tensor product by $\otimes_{L_1, L_2}^{L_3}$ which combines a type-$L_1$ vector $f^{(L_1)}$ and a type-$L_2$ vector $g^{(L_2)}$ to produce a

type-$L_3$ vector $h^{(L_3)}$ via Clebsch–Gordan coefficients:

$$h_{m_3}^{(L_3)} = (f^{(L_1)} \otimes_{L_1,L_2}^{L_3} g^{(L_2)})_{m_3}$$

$$= \sum_{m_1=-L_1}^{L_1} \sum_{m_2=-L_2}^{L_2} C_{(L_1,m_1)(L_2,m_2)}^{(L_3,m_3)} f_{m_1}^{(L_1)} g_{m_2}^{(L_2)} \quad (1)$$

where order $m_1$ refers to the $m_1$-th component of $f^{(L_1)}$. The Clebsch-Gordan coefficients $C_{(L_1,m_1)(L_2,m_2)}^{(L_3,m_3)}$ are non-zero only when $|L_1 - L_2| \le L_3 \le |L_1 + L_2|$, which constrains the possible degrees of the output $h^{(L_3)}$. In practice, we restrict the maximum degree to a fixed hyper-parameter $L_{\max}$ and discard components with $L > L_{\max}$ to control computational cost and prevent the feature dimensionality from growing without bound. Many equivariant GNNs implement message passing through equivariant convolutions, which perform tensor products between input irreps features $x^{(L_1)}$ and spherical harmonics of relative position vectors $Y^{(L_2)}(\frac{\vec{r}}{||\vec{r}||})$. This construction enables the network to combine geometric information with learned features while strictly preserving $SE(3)$-equivariance.

### 2.2. Equiformer Series

Equiformer (Liao & Smidt, 2023) introduced three key modifications that adapt the Transformer architecture (Vaswani et al., 2017) into a powerful $E(3)/SE(3)$-equivariant GNN. First, scalar features are replaced with equivariant irreps features, enabling the network to represent and propagate symmetry-aware information. Second, all core Transformer operations are generalized to their equivariant counterparts. This includes tensor products, as well as equivariant versions of linear layers, layer normalization, and nonlinearities such as gate activations (Weiler et al., 2018). Third, Equiformer applies nonlinear functions to both attention weights and value vectors during message passing, extending standard dot-product attention to improve its expressivity (Brody et al., 2022).

While Equiformer successfully generalizes Transformers to 3D atomistic modeling, the computational cost of tensor product operations limits the maximum degree of equivariant representations that could be used in practice, thereby constraining the overall expressivity (Joshi et al., 2023). To address this limitation, EquiformerV2 (Liao et al., 2024b) adopts eSCN convolutions (Passaro & Zitnick, 2023), which significantly reduce the computational complexity of tensor products and enable the efficient use of higher-degree representations. We refer readers to Passaro & Zitnick (2023) for mathematical details. In addition, EquiformerV2 introduced three architectural improvements for better scaling to higher degrees. First, attention re-normalization is proposed, adding an extra layer normalization to the nonlinear transformations of attention weights. Second, separable $S^2$ activation, based on the original $S^2$ activation (Cohen et al.,

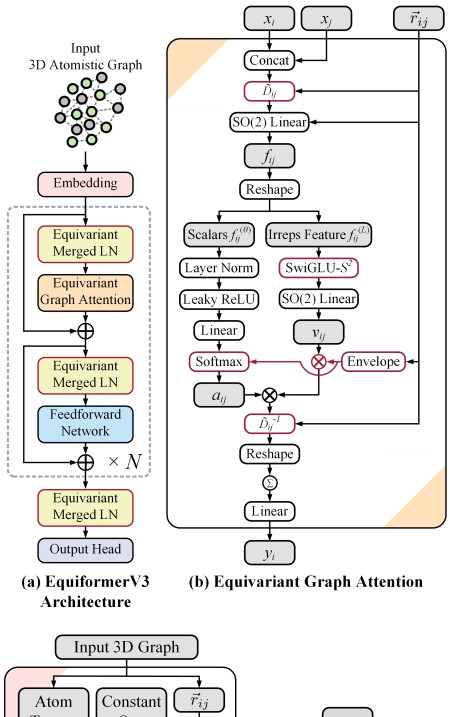

**(a) EquiformerV3 Architecture**   **(b) Equivariant Graph Attention**

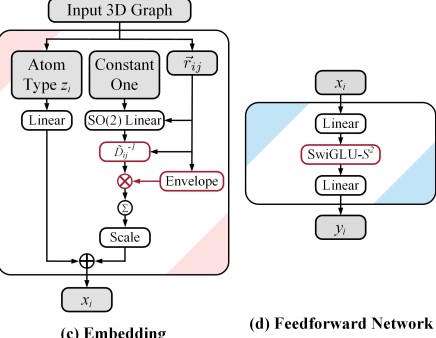

**(c) Embedding**   **(d) Feedforward Network**

*Figure 1.* EquiformerV3 architecture. The proposed improvements are highlighted in red. We encode input 3D atomistic graphs with atom and edge-degree embeddings and process with Transformer blocks, which consist of equivariant merged layer normalization (LN), equivariant graph attention and feedforward networks.

2018), is introduced for better mixing information across degrees. Third, separable layer normalization is proposed to preserve the relative importance of equivariant features for better generalization.

## 3. EquiformerV3

Starting from EquiformerV2 (Liao et al., 2024b), we describe the improvements leading to EquiformerV3 in Section 3.1, Section 3.2 and Section 3.3. For completeness, we present the overall network architecture in Section B. Figure 1 illustrates the EquiformerV3 architecture and highlights the proposed improvements.

### 3.1. Optimizing Software Implementation

We improve the implementation of EquiformerV2 by fusing redundant operations and enabling compilation, resulting in $1.75\times$ speedup in training on the OC20 dataset (Chanussot* et al., 2021) while maintaining the same accuracy.

**Fusing Redundant Operations.** EquiformerV2 adopts eSCN convolutions (Passaro & Zitnick, 2023), which decompose the tensor product between irreps features $x$ and spherical harmonics of relative position vectors $\vec{r}_{ij}$ into two steps: (1) applying rotation matrices $D_{ij}$ derived from $\vec{r}_{ij}$, and (2) applying $SO(2)$ linear layers to the rotated features $D_{ij}x$. In the original implementation, the $SO(2)$ linear layers additionally require multiplication by permutation matrices $S$ to rearrange the ordering of degree $L$ and order $m$ in $D_{ij}x$, which introduces redundant computation. We observe that the permutation can be fused directly into the rotation matrix by defining $\widetilde{D}_{ij} = S \cdot D_{ij}$. Then, $SO(2)\_Linear(S \cdot (D_{ij}x)) = SO(2)\_Linear((S \cdot D_{ij})x) = SO(2)\_Linear(\widetilde{D}_{ij}x)$. By permuting $D_{ij}$ once to form $\widetilde{D}_{ij}$, all subsequent permutation operations in $SO(2)$ linear layers can be eliminated. We note that writing customized CUDA kernels for eSCN convolutions, in a similar manner to libraries like cuEquivariance (Geiger et al., 2024) and OpenEquivariance (Bharadwaj et al., 2025), could further improve efficiency in future work.

**Enabling Compilation.** We fix several implementation issues to enable compilation (i.e., `torch.compile()`) with dynamic shapes. These changes mainly include pre-computing constant tensors, such as the permutation matrices used in $SO(2)$ linear layers, and explicitly specifying the output shapes of scatter operations (e.g., `torch_scatter.scatter(dim_size=num_nodes)`) instead of relying on the default behavior of inferring shapes from index tensors.

### 3.2. Simple and Effective Modifications to EquiformerV2

**Equivariant Merged Layer Normalization.** The equivariant layer normalization used by Equiformer (Liao & Smidt, 2023) normalizes vectors of each degree $L$ independently. As a result, the average magnitudes of different degrees become the same after the normalization, which removes the relative importance between different degrees, and can negatively affect training dynamics. To address this issue, EquiformerV2 (Liao et al., 2024b) introduces separable layer normalization, which applies two separate normalizations: one for scalar features ($L = 0$) and one shared across all higher-degree features ($L > 0$). This modification preserves the relative magnitudes of non-scalar degrees ($L > 0$) and improves the empirical performance. In EquiformerV3, we further refine this idea by adopting an equivariant merged layer normalization, which uses a single, shared normalization for all degrees $L \geq 0$. Mathematically, let $x \in \mathbb{R}^{(L_{max}+1)^2 \times C}$ be an input irreps feature of maximum degree $L_{max}$ and $C$ channels, and we use $x_{m,i}^{(L)}$ to denote the $L$-th degree, $m$-th order and $i$-th channel. We first calculate the root mean square (RMS) values $\sigma^{(L)}$ for each degree $L$. For $L = 0$, $\sigma^{(0)} = \sqrt{\frac{1}{C}\sum_{i=1}^{C}(x_{0,i}^{(0)} - \mu^{(0)})^2}$,

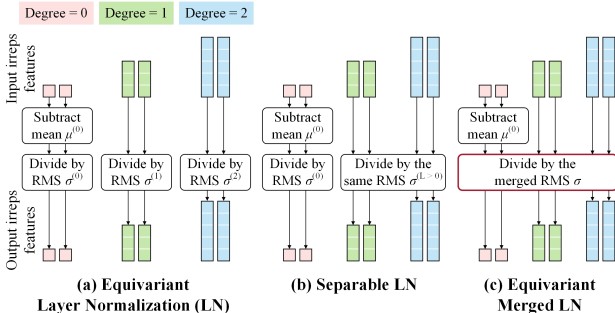

Degree = 0  Degree = 1  Degree = 2

(a) Equivariant Layer Normalization (LN)

(b) Separable LN

(c) Equivariant Merged LN

*Figure 2.* Illustration of how different normalizations calculate statistics. In the proposed equivariant merged layer normalization, we share the merged RMS across all degrees as highlighted in red.

where the mean $\mu^{(0)} = \frac{1}{C}\sum_{i=1}^{C}x_{0,i}^{(0)}$. For $L > 0$, $\sigma^{(L)} = \sqrt{\frac{1}{C}\sum_{i=1}^{C}\frac{1}{2L+1}\sum_{m=-L}^{L}\left(x_{m,i}^{(L)}\right)^2}$. We then sum over $L$ to obtain the shared, merged RMS value $\sigma = \sqrt{\frac{1}{L_{max}+1}\sum_{L=0}^{L_{max}}\left(\sigma^{(L)}\right)^2}$. All degrees are normalized using this shared $\sigma$. The outputs $y$ are given by $y^{(0)} = \gamma^{(0)} \circ \left(\frac{x^{(0)} - \mu^{(0)}}{\sigma}\right) + \beta^{(0)}$ and $y^{(L)} = \gamma^{(L)} \circ \left(\frac{x^{(L)}}{\sigma}\right)$ for $L > 0$, where $\gamma^{(0)}, \gamma^{(L)}, \beta^{(0)} \in \mathbb{R}^C$ are learnable parameters for affine transformation. Figure 2 compares these three normalizations. The key difference lies in how the RMS values in the divisor are obtained.

**Better Architectural Hyper-Parameters for Feedforward Networks.** In equivariant GNNs, the primary bottleneck arises from the tensor product operations applied to edge features, which are both compute- and memory-intensive. This step limits the practical size and capacity of edge-wise networks. In contrast, node-wise components such as the feedforward networks (FFN) incur significantly lower computational and memory costs, making them an efficient lever for increasing model capacity. Motivated by this inefficiency, we increase the hidden size of FFNs by $4\times$, improving model capacity and performance with minimal additional computational overhead.

**Attention with Smooth Radius Cutoff.** Previous generations of Equiformer were primarily designed to approxmiate energy and forces from single-point DFT calculations. For tasks that require smoothly varying potential energy surface (PES), however, continuity with respect to atomic positions is essential. Following eSEN (Fu et al., 2025), we introduce envelope functions (Gasteiger et al., 2020) into the attention mechanism to enforce a smooth radius cutoff. In the attention module, the message $m_{ij}$ sent from node (or atom) $j$ to node $i$ is given by

$$m_{ij} = a_{ij} \times v_{ij}, \tag{2}$$

where attention weights $a_{ij}$ are scalars (or type-$0$ vectors) and value vectors $v_{ij}$ are equivariant irreps features containing degrees from $0$ to $L_{max}$. The attention weights $a_{ij}$

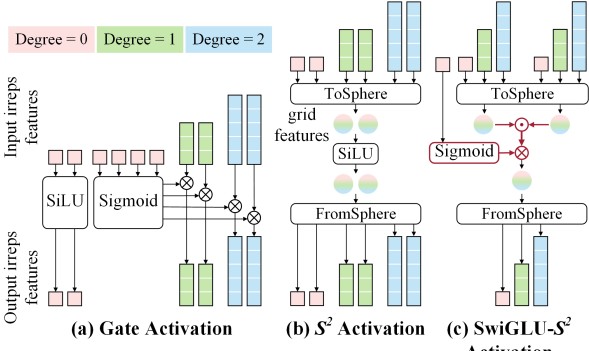

*Figure 3.* Illustration of different activation functions. The colorful circles are grid features. One circle represents one channel, which contains $R_\phi \times R_\theta$ grid points on $S^2$. In the proposed SwiGLU-$S^2$ activation, we apply both nonlinearity and multiplication to grid features as highlighted in red.

are computed by applying a softmax operation to attention logits $z_{ij}$ over the neighborhood $\mathcal{N}(i)$:

$$a_{ij} = \text{softmax}_j(z_{ij}) = \frac{\exp(z_{ij})}{\sum_{k \in \mathcal{N}(i)} \exp(z_{ik})} \quad (3)$$

Applying envelope functions only to the messages $m_{ij}$, as done in eSEN, however, is insufficient to guarantee smooth predictions because the softmax operation depends on all neighbors in $\mathcal{N}(i)$. When atoms enter or leave the cutoff radius, the denominator changes abruptly, leading to discontinuities in $a_{ij}$. We resolve this by incorporating an additional envelope function directly into the softmax operation. The resulting attention with smooth radius cutoff is:

$$
\begin{aligned}
a_{ij} &= \text{softmax}_j(z_{ij}) \\
&= \frac{\text{envelope}(||\vec{r}_{ij}||) \cdot \exp(z_{ij})}{\sum_{k \in \mathcal{N}(i)} \text{envelope}(||\vec{r}_{ik}||) \cdot \exp(z_{ik})}, \quad (4) \\
m_{ij} &= a_{ij} \times (\text{envelope}(||\vec{r}_{ij}||) \cdot v_{ij})
\end{aligned}
$$

where $\text{envelope}(||\vec{r}_{ij}||)$ denotes the envelope function parametrized by relative distance $||\vec{r}_{ij}||$.

### 3.3. SwiGLU-$S^2$ Activation

We incorporate $S^2$ activation and fast tensor product operations based on projection onto the unit sphere $S^2$ to propose stronger SwiGLU-$S^2$ activation. A comparison between different activation functions is shown in Figure 3.

$S^2$ **Activation.** First proposed in Spherical CNNs (Cohen et al., 2018), $S^2$ activation projects irreps features onto the unit sphere $S^2$, applies standard (unconstrained) activation functions to the projection, and then projects the result back to irreps space. Mathematically, we consider an input irreps feature $x \in \mathbb{R}^{(L_{max}+1)^2}$ with maximum degree $L_{max}$ and a single channel (the same process is applied independently to each channel). We use $\phi$ and $\theta$ to denote the longitude and the latitude of $S^2$. The output $y$ of the $S^2$ activation function is as follows:

$$
\begin{aligned}
x^{grid}(\phi, \theta) &\leftarrow \text{ToSphere}(x, \phi, \theta) \\
&= \sum_{L=0}^{L_{max}} \sum_{m=-L}^{L} Y_m^{(L)}(\phi, \theta) \cdot x_m^{(L)} \\
y^{grid}(\phi, \theta) &\leftarrow F(x^{grid}(\phi, \theta)) \\
y_m^{(L)} &\leftarrow \text{FromSphere}(y^{grid}(\phi, \theta), L, m) \\
&= \int_{\phi=0}^{2\pi} \int_{\theta=0}^{\pi} y^{grid}(\phi, \theta) Y_m^{(L)}(\phi, \theta) \sin\theta \, d\theta \, d\phi
\end{aligned}
$$
(5)

ToSphere() denotes projecting onto $S^2$ by taking the inner product between spherical harmonics $Y_m^{(L)}$ and $x_m^{(L)}$. FromSphere() denotes the inverse of ToSphere() and converts $y^{grid}$ back to irreps feature $y$. In practice, both integrals are approximated by sampling $R_\phi$ and $R_\theta$ grid points along $\phi$ and $\theta$, so the grid features $x^{grid}, y^{grid}$ have size $R_\phi \times R_\theta$. The choice of nonlinear function $F$ influences how dense this sampling must be to maintain low equivariance error. Previous works (Zitnick et al., 2022; Passaro & Zitnick, 2023; Liao et al., 2024b) typically choose SiLU (Elfwing et al., 2017; Ramachandran et al., 2017) as $F$.

**Fast Tensor Product Operations Based on Projection onto the Unit Sphere** $S^2$**.** Tensor products can be efficiently computed by projecting irrep features to the Fourier space or onto the unit sphere $S^2$ (Xin et al., 2021; Luo et al., 2024; Xie et al., 2025). This construction does not compute the full tensor product, but only the symmetry-respecting paths corresponding to the symmetric components used in equivariant feature interactions, avoiding the cost of explicitly forming all Clebsch–Gordan couplings (Xie et al., 2025). Focusing on the $S^2$ formulation, which is directly compatible with $S^2$ activation, we use elementwise multiplication between grid features to compute the tensor products between two irreps features $x, y \in \mathbb{R}^{(L_{max}+1)^2}$:

$$
\begin{aligned}
x^{grid}(\phi, \theta) &\leftarrow \text{ToSphere}(x, \phi, \theta) \\
y^{grid}(\phi, \theta) &\leftarrow \text{ToSphere}(y, \phi, \theta) \\
z^{grid}(\phi, \theta) &\leftarrow x^{grid}(\phi, \theta) \odot y^{grid}(\phi, \theta) \\
z_m^{(L)} &\leftarrow \text{FromSphere}(z^{grid}(\phi, \theta), L, m)
\end{aligned}
$$
(6)

By appropriately modifying the Clebsch-Gordan coefficients, this procedure recovers the desired symmetric paths of the tensor product $z = x \otimes y$. Both ToSphere() and FromSphere() have the same complexity $\mathcal{O}(L_{max}^4)$, and the elementwise multiplication has the compleixty $\mathcal{O}(L_{max}^2)$. Therefore, Equation 6 reduces the complexity of $x \otimes y$ from $\mathcal{O}(L_{max}^6)$ to $\mathcal{O}(L_{max}^4)$. We refer readers to Xie et al. (2025) for more details about fast tensor product operations and on which paths are included in tensor products on the sphere (symmetric versus antisymmetric).

**Proposed Stronger SwiGLU-$S^2$ Activation.** We propose a stronger SwiGLU-$S^2$ activation function that applies both

nonlinearity and multiplication to grid features as follows:

$$
\begin{aligned}
&\text{SwiGLU-}S^2(x_{scalar}, x_1^{grid}, x_2^{grid}) \\
&= \text{Sigmoid}(x_{scalar}) \cdot x_1^{grid} \odot x_2^{grid}
\end{aligned}
\tag{7}
$$

where $x_{scalar} \in \mathbb{R}^C$ is a scalar feature of $C$ channels, $x_1^{grid}, x_2^{grid} \in \mathbb{R}^{R_\phi \times R_\theta \times C}$ are two different grid features. Compared to directly applying SwiGLU (Shazeer, 2020) to grid features (i.e., $\text{SiLU}(x_1^{grid}) \odot x_2^{grid} = \text{Sigmoid}(x_1^{grid}) \odot x_1^{grid} \odot x_2^{grid}$), we find that using $x_{scalar}$ as the nonlinear gating instead of $x_1^{grid}$ can achieve similar empirical results. We attribute this to the multiplication interaction with $x_2^{grid}$, while enabling strict equivariance through scalar-only non-linear gating. Relative to SiLU-based $S^2$ activation used in EquiformerV2, SwiGLU-style gating has consistently improved performance in large language models (Chowdhery et al., 2022; Grattafiori et al., 2024; DeepSeek-AI et al., 2025; OpenAI et al., 2025). Moreover, $S^2$ activation itself outperforms (Liao et al., 2024b) gate activation (Weiler et al., 2018) used in equivariant models like Equiformer (Liao & Smidt, 2023), eSEN (Fu et al., 2025) and UMA (Wood et al., 2025). Together, these motivate SwiGLU-$S^2$ as a strictly equivariant and more expressive activation than those used in prior equivariant GNNs.

**Many-Body Interactions for Better Theoretical Expressivity.** The multiplication in Equation 7 (i.e., $x_1^{grid} \odot x_2^{grid}$) is equivalent to self tensor products in irreps space (i.e., $x \otimes x$). As shown in Joshi et al. (2023), stacking self tensor products can incorporate many-body interactions enabling stronger local descriptors and allowing a single message-passing step to distinguish geometric graphs that require higher body-order scalarization. Since the discriminative power is linked with the theoretical expressivity of GNNs (Xu et al., 2019; Morris et al., 2021), incorporating many-body interactions through SwiGLU-$S^2$ activations directly improves the expressivity of equivariant GNNs. We reproduce the body order experiment of Joshi et al. (2023) in Section C and show that the SwiGLU-$S^2$ activation can distinguish all tested geometric graphs, whereas other activations cannot. Importantly, this perspective suggests that the empirical success of SwiGLU in other domains may also stem from its ability to introduce higher-order interactions.

**Strict Equivariance with Reduced Complexity of Sampling $S^2$ Grids.** The original $S^2$ activation directly applies activation to grid features, which can introduce high-frequency components and sampling errors, thereby breaking strict equivariance and energy conservation (Fu et al., 2025). Although increasing the number of grid points can reduce this error, maintaining strict equivariance in this way leads to prohibitively large grid sizes and computational cost. In contrast, SwiGLU-$S^2$ applies activation to only scalars and uses bilinear multiplications on grid features.

This structure avoids injecting high-frequency content into the spherical grid and significantly reduces the number of grid points required to preserve strict equivariance. We compare the equivariance errors of $S^2$ and SwiGLU-$S^2$ activations with different degrees and numbers of grid points in Section D. In particular, for $L_{max} = 6$, EquiformerV3 maintains strict equivariance while reducing the number of grid points in attention by $50.6\%$. Moreover, combining strict equivariance and attention using a smooth radius cutoff enables accurate modeling of smoothly varying PES.

## 4. Experiments

### 4.1. OC20

**Dataset and Task.** The large and diverse Open Catalyst 2020 dataset (OC20) (Chanussot* et al., 2021) consists of about 1.2M Density Functional Theory (DFT) relaxation trajectories computed with the revised Perdew-Burke-Ernzerhof (RPBE) functional (Hammer et al., 1999). Each trajectory starts from an initial structure of an adsorbate molecule placed on a catalyst surface and is then relaxed to a local energy minimum. The primary task in OC20 is Structure to Energy and Forces (S2EF), which is to predict the energy and per-atom forces given a structure from the trajectories. These predictions are evaluated on energy and force mean absolute error (MAE).

**Training Details.** Please refer to Section E for details on architectures and hyper-parameters.

**Results.** We conduct ablation studies on the proposed improvements and summarize the results in Table 1. We start with the EquiformerV2 (Liao et al., 2024b) model with $N = 8$ Transformer blocks and $L_{max} = 6$. We adopt most of the setting of EquiformerV2 except that we predict total energy instead of adsorption energy (Index 2) by following the practices in Abdelmaqsoud et al. (2024); Wood et al. (2025). This approach improves accuracy because total energy models can better predict ill-posed adsorption energies caused by surface reconstructions (Abdelmaqsoud et al., 2024), thus enabling continuous improvements in energy errors. First, optimizing the software implementation achieves $1.75\times$ speedup, reducing the training time from 270 GPU-hours to 154, while maintaining similar errors (Index 3). Second, using the merged RMS value for all degrees as in equivariant merged layer normalization improves energy and force MAE (Index 4). Third, the improved architectural hyper-parameters for feedforward networks (FFNs) increases model sizes by $22\%$ for better performance while increasing training time by only $8.6\%$ (Index 5). Fourth, incorporating smooth radius cutoff to attention has neutral effects on learning single-point energy and forces under the setting of direct prediction (Index 6) while it can help learning smoothly varying PES and energy conservation (Fu et al., 2025). Finally, using the proposed SwiGLU-$S^2$ ac-

| Index | Method | Energy | Force | Number of parameters | Training time (GPU-hours) |
|---|---|---|---|---|---|
| 1 | EquiformerV2 baseline | 296 | 21.23 | 54M | 270 |
| 2 | + Predicting total energy | 242 | 19.73 | 54M | 270 |
| 3 | + Better implementation | 242 | 19.73 | 54M | 154 |
| 4 | + Equivariant merged LN | 236 | 19.28 | 54M | 150 |
| 5 | + Improved FFN hyper-parameters | 209 | 18.96 | 66M | 163 |
| 6 | + Smooth attention cutoff | 213 | 18.82 | 66M | 163 |
| 7 | + SwiGLU-$S^2$ activation | 201 | 18.15 | 91M | 171 |

*Table 1.* Ablation studies on EquiformerV3 architecture. All models are trained on the 2M split of the OC20 S2EF dataset, and errors are averaged over the four validation sub-splits. We report mean absolute errors (MAE) for forces in meV/Å and energy in meV, and lower is better. Training time is measured on H100 GPUs. "LN" denotes layer normalization.

tivation can further reduce energy and force MAE (Index 7). Since the SwiGLU-$S^2$ activation halves the number of channels, we increase the number of input channels by $2\times$ so that subsequent layers remain the same. This results in 37.9% additional learnable parameters when using the SwiGLU-$S^2$ activation. Moreover, as we do not directly apply activation functions to grid features, we can adjust the number of grid points for better efficiency while preserving strict equivariance. Specifically, for $L_{max} = 6$, the original $S^2$ activation in EquiformerV2 uses $(R_\phi, R_\theta) = (18, 18)$ and thus 324 grid points in both attention and FFNs. As for the SwiGLU-$S^2$ activation, based on the equivariance errors in Section D, we use $(R_\phi, R_\theta) = (8, 20)$ in attention and $(R_\phi, R_\theta) = (20, 20)$ in FFNs for strict equivariance. Reducing the number of grid points from 324 to 160 in attention enables using more parameters while maintaining similar training time. Together, the proposed improvements decrease energy MAE by 41 meV and force MAE by 1.58 meV/Å and saves $1.58\times$ training time (Index 2 versus Index 7). The gain is therefore comparable to that brought by the architectural improvements of EquiformerV2, which reduces force MAE by 1.4 meV/Å but increases training time by $1.46\times$ (Index 1 versus Index 5 in Table 1(a) in Liao et al. (2024b)). Additionally, comparing EquiformerV2 with $1.5\times$ more blocks and trained for $2.5\times$ more epochs, which has force MAE of 19.4 meV/Å (Table 1(b) in Liao et al. (2024b)), we find that EquiformerV3 can achieve similar force MAE and save $5.9\times$ ($= \frac{270}{171} \times 1.5 \times 2.5$) training time.

### 4.2. OMat24 Dataset

**Dataset and Task.** The Open Materials 2024 dataset (OMat24) (Barroso-Luque et al., 2024) samples relaxed structures from Alexandria dataset (Schmidt et al., 2024) and generates more than 110M diverse non-equilibrium crystal structures with Boltzmann sampling of rattled structures, ab initio molecular dynamics, and relaxations of rattled structures. Energies, forces and stresses are calculated with the Perdew-Burke-Ernzerhof density functional (Perdew et al., 1996) in the Vienna ab initio Simulation Package (VASP) (Kresse & Furthmüller, 1996), following the default setting of Materials Project (Jain et al., 2013) but with some

| Index | Model | Prediction | Energy | Force | Stress | Number of parameters |
|---|---|---|---|---|---|---|
| | EquiformerV2-S | Direct | 11 | 49.2 | 2.4 | 31M |
| | EquiformerV2-M | Direct | 10 | 44.8 | 2.3 | 87M |
| | EquiformerV2-L | Direct | 9.6 | 43.1 | 2.3 | 154M |
| | eSEN | Gradient | 10.7 | 47.3 | 2.6 | 30M |
| | UMA-S | Gradient | 11.3 | 57.1 | 2.9 | 6M |
| | UMA-M | Gradient | 10.0 | 47.3 | 2.7 | 50M |
| | UMA-L | Direct | 9.7 | 43.5 | 2.5 | 700M |
| | EquiformerV3 | | | | | |
| 1 | $L_{max} = 4$ | Direct | 10.5 | 45.7 | 2.7 | 34M |
| 2 | $L_{max} = 4$ + grad. ft. | Gradient | 10.4 | 43.5 | 2.6 | 30M |
| 3 | $L_{max} = 6$ | Direct | 9.8 | 43.1 | 2.6 | 57M |
| 4 | $L_{max} = 6$ + grad. ft. | Gradient | 10.1 | 41.6 | 2.5 | 49M |

*Table 2.* Results on the OMat24 validation set. We report mean absolute errors (MAE) for energy in meV per atom, forces in meV/Å and stress in meV/Å$^3$, and lower is better. "Prediction" denotes whether a model uses "Direct" or "Gradient" methods to predict forces and stress. "grad. ft." denotes gradient fine-tuning.

exceptions. Here we focus on predicting energy, forces and stress given a structure, and the evaluation metric is MAE.

**Training Details.** Please refer to Section F for details on architectures, hyper-parameters and training time.

**Results.** The comparison on the validation set is summarized in Table 2. We train EquiformerV3 with $L_{max} = 4$ and 6 and follow the practice of direct pre-training and gradient fine-tuning by eSEN (Fu et al., 2025). Index 1 and Index 3 are trained with direct methods. Index 2 and Index 4 start with model weights from Index 1 and Index 3, respectively, and are then fine-tuned with gradient methods. Since we additionally use regularizations such as dropout (Srivastava et al., 2014) and stochastic depth (Huang et al., 2016), as well as the auxiliary task of DeNS (Liao et al., 2024a), during direct pre-training, models trained with direct methods fit the training data less well compared to models trained with gradient methods. Moreover, because the training and validation sets are in-distribution, this explains why models trained with gradient methods (Index 2 and Index 4) perform better. Compared to EquiformerV2 (Liao et al., 2024b), EquiformerV3 with $L_{max} = 4$ (Index 2) achieves comparable force MAE while being $5\times$ smaller in model sizes. The differences in energy and stress MAE may arise because labels are normalized differently in direct and gradient methods. Specifically, when using gradient methods, we normalize energy and stress labels with the statistics of forces instead of those of energies and stresses. Compared to eSEN and UMA (Wood et al., 2025) under the same setting of gradient methods, EquiformerV3 with $L_{max} = 4$ (Index 2) achieves overall better performance. Moreover, EquiformerV3 with $L_{max} = 4$ (Index 2) has similar results to UMA-L and is $23\times$ smaller. When increasing $L_{max}$ to 6 (Index 4), EquiformerV3 further improves upon previous models. Finally, since the validation set is in-distribution with respect to the training set and single-point errors do not fully reflect how ML models are used, we further compare different models when fine-tuned for Matbench Discovery (Riebesell et al., 2024) in Section 4.3.

| Model | Prediction | F1 ↑ | RMSD ↓ | $\kappa_{SRME}$ ↓ | CPS ↑ |
|---|---|---|---|---|---|
| MACE-MP-0 | Gradient | 0.669 | 0.091 | 0.682 | 0.637 |
| ORB v2 MPtrj | Direct | 0.765 | 0.101 | 1.726 | 0.470 |
| EquiformerV2[†‡] | Direct | 0.815 | 0.076 | 1.676 | 0.522 |
| GRACE-2L-MPtrj | Gradient | 0.691 | 0.090 | 0.525 | 0.681 |
| SevenNet-l3i5 | Gradient | 0.760 | 0.085 | 0.550 | 0.714 |
| eSEN-30M-MP[‡] | Gradient | 0.831 | 0.075 | 0.340 | 0.797 |
| Eqnorm MPtrj | Gradient | 0.786 | 0.084 | 0.408 | 0.756 |
| DPA-3.1-MPtrj | Gradient | 0.803 | 0.080 | 0.650 | 0.718 |
| Nequix MP | Gradient | 0.751 | 0.085 | 0.446 | 0.729 |
| NequIP-MP-L | Gradient | 0.761 | 0.086 | 0.452 | 0.733 |
| MatRIS-10M-MP | Gradient | 0.847 | 0.072 | 0.489 | 0.778 |
| EquiformerV3[‡] | Gradient | **0.863** | **0.070** | **0.275** | **0.830** |

*Table 3.* Matbench Discovery results of compliant models trained on only MPtrj. "Prediction" denotes whether a model uses "Direct" or "Gradient" methods to predict forces and stress. † corresponds to "eqV2 S DeNS" on the leaderboard. ‡ denotes training with the auxiliary task of DeNS (Liao et al., 2024a).

### 4.3. Matbench Discovery

**Benchmark Details.** Matbench Discovery (Riebesell et al., 2024) is a public leaderboard and evaluation framework that ranks ML models based on tasks related to real discovery workflows of inorganic crystalline structures. The primary task involves predicting thermodynamic stability of crystal structures through geometry optimization and energy prediciton. Key metrics include F1 score, which measures precision and recall in stability classification, and root mean square deviation (RMSD) between the ML-predicted relaxed structures and DFT-relaxed reference structures. Additionally, the thermal conductivity task examines individual harmonic and anharmonic phonon mode contributions to thermal conductivity based on the Wigner formulation of heat transport (Simoncelli et al., 2022), evaluating ML models on the accuracy of higher-order derivatives of their learned PES. After relaxing structures, second- and third-order force constants are calculated with the supercell method to determine symmetric relative mean error in predicting thermal conductivity ($\kappa_{SRME}$) as proposed by Póta et al. (2024). Matbench Discovery defines a combined performance score (CPS) that weighs F1, RMSD, and $\kappa_{SRME}$ together to reflect holistic model performance.

**Training Details.** Please refer to Section G for details on architectures, hyper-parameters and training time.

**Results.** We consider the settings of compliant models trained on only MPtrj dataset (Deng et al., 2023) and non-compliant models trained on additional datasets like OMat24 (Barroso-Luque et al., 2024). We follow the practice of direct pre-training and gradient fine-tuning by eSEN (Fu et al., 2025). During direct pre-training, we use the auxiliary task of DeNS (Liao et al., 2024a). The comparison between models trained on MPtrj is summarized in Table 3. EquiformerV3 achieves the best results on all metrics. Compared to EquiformerV2, which achieves a strong F1 score but fails the thermal conductivity task with high $\kappa_{SRME}$, EquiformerV3 significantly reduces $\kappa_{SRME}$ from 1.676 to 0.275. This improvement indicates that

| Model | Prediction | F1 ↑ | RMSD ↓ | $\kappa_{SRME}$ ↓ | CPS ↑ | Training time (GPU-hours) |
|---|---|---|---|---|---|---|
| ORB v2 | Direct | 0.880 | 0.097 | 1.734 | 0.528 | - |
| EquiformerV2[†] | Direct | 0.917 | 0.069 | 1.771 | 0.558 | - |
| MACE-MPA-0 | Gradient | 0.852 | 0.073 | 0.412 | 0.795 | - |
| MatterSim v1 5M | Gradient | 0.862 | 0.073 | 0.575 | 0.767 | - |
| eSEN-30M-OAM | Gradient | 0.925 | 0.061 | 0.170 | 0.888 | - |
| ORB v3 | Gradient | 0.905 | 0.075 | 0.210 | 0.860 | - |
| EquFlash | Gradient | 0.919 | **0.060** | 0.158 | 0.888 | - |
| UMA-M | Gradient | 0.930 | 0.061 | 0.195 | 0.885 | >129k |
| NequIP-OAM-XL | Gradient | 0.906 | 0.063 | 0.125 | 0.886 | - |
| PET-OAM-XL | Gradient | 0.924 | **0.060** | 0.119 | 0.898 | 20.5k |
| EquiformerV3[‡] | Gradient | **0.931** | **0.060** | **0.118** | **0.902** | 5.7k |

*Table 4.* Matbench Discovery results of non-compliant models trained on larger datasets like OMat24. "Prediction" denotes whether a model uses "Direct" or "Gradient" methods to predict forces and stress. † corresponds to "eqV2 M" on the leaderboard. ‡ denotes training with the auxiliary task of DeNS. The training time of UMA-M is measured on H200 GPUs, and others use H100.

attention with smooth cutoff and the SwiGLU-$S^2$ activation, together with gradient methods, enable EquiformerV3 to generalize well to tasks requiring higher-order derivatives of PES. Compared to eSEN, EquiformerV3 shows that additional architectural improvements can generally improve the performance on all the tasks and particularly decrease $\kappa_{SRME}$ by 19%. We also compare EquiformerV3 with models trained on larger datasets in Table 4. Following previous works, we pre-train EquiformerV3 on OMat24 and fine-tune (Index 2 in Table 2) on the MPtrj and sub-sampled Alexandria (Barroso-Luque et al., 2024; Schmidt et al., 2024) datasets with gradient methods. EquiformerV3 with $L_{max} = 4$ has better results on all metrics and becomes the first to achieve CPS > 0.9. Compared to eSEN, EquiformerV3 improves $\kappa_{SRME}$ by 31% and has a higher F1 score. Although the absolute improvement in F1 scores might seem small, it exceeds the gain obtained by increasing the pre-training dataset size by more that $4\times$ as done by UMA-M (Wood et al., 2025), indicating a meaningful improvement in performance. Compared to UMA-M, EquiformerV3 consistently achieves better results on all metrics and saves $22.6\times$ training time. While NequIP-OAM-XL (Batzner et al., 2022; Tan et al., 2025) achieves comparable $\kappa_{SRME}$, EquiformerV3 achieves a better F1 score and reduces the training epochs on OMat24 from 30 to 6. Compared to PET-OAM-XL (Mazitov et al., 2025; Bigi et al., 2026), EquiformerV3 has a higher F1 score and comparable $\kappa_{SRME}$ while reducing the number of training epochs from 15 to 6 and saving $3.6\times$ training time.

## 5. Conclusion

In this work, we introduce EquiformerV3 to enhance the efficiency, expressivity, and generality of $SE(3)$-equivariant graph attention Transformers. Building on EquiformerV2, we have three key advances: optimizing software implementation, simple and effective modifications to EquiformerV2, and SwiGLU-$S^2$ activation. With these improvements, EquiformerV3 achieves state-of-the-art results on OC20, OMat24 and Matbench Discovery.

## Impact Statement

EquiformerV3 enables more accurate approximation of quantum mechanical calculations for accelerating applications in chemistry and material science. We hope these promising results will encourage the community to make further progress rather than use these methods for adversarial purposes. We note that these methods only facilitate the identification of molecules or materials with specific properties and that they require substantial efforts to synthesize and deploy at scale.

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

## Appendix

## A. Related Works

$SE(3)/E(3)$-equivariant neural networks (Thomas et al., 2018; Kondor et al., 2018; Weiler et al., 2018; Fuchs et al., 2020; Miller et al., 2020; Townshend et al., 2020; Batzner et al., 2022; Jing et al., 2021; Schütt et al., 2021; Satorras et al., 2021; Unke et al., 2021; Brandstetter et al., 2022; Thölke & Fabritiis, 2022; Le et al., 2022; Musaelian et al., 2022; Batatia et al., 2022; Liao & Smidt, 2023; Passaro & Zitnick, 2023; Liao et al., 2024b; Park et al., 2024; Fu et al., 2025; Wood et al., 2025) operate on equivariant irreps features built from vector spaces of irreducible representations (irreps) to achieve equivariance to 3D rotation (Thomas et al., 2018; Weiler et al., 2018; Kondor et al., 2018). Previous works on equivariant networks mainly differ in which equivariant operations they use and how they combine those operations. Tensor Field Networks (Thomas et al., 2018) and NequIP (Batzner et al., 2022) use tensor products to build equivariant graph convolution with linear messages. NequIP additionally utilizes equivariant gate activation (Weiler et al., 2018) for node-wise functions. SEGNN (Brandstetter et al., 2022) applies gate activation to edge features for non-linear messages (Gilmer et al., 2017; Sanchez-Gonzalez et al., 2020), and the non-linear messages outperform linear messages. SE(3)-Transformer (Fuchs et al., 2020) adopts equivariant dot product attention with linear messages. Equiformer(V1) (Liao & Smidt, 2023) improves upon the above models by combining MLP attention and non-linear messages and introducing equivariant layer normalization. However, these networks rely on compute-intensive $SO(3)$ tensor products during message passing, and therefore they are limited to small values for maximum degrees $L_{max}$ of equivariant representations. SCN (Zitnick et al., 2022) proposes rotating irreps features based on relative position vectors and identifies a subset of spherical harmonics coefficients to which they can apply unconstrained functions while minimizing equivariance errors. eSCN (Passaro & Zitnick, 2023) improves upon SCN by replacing unconstrained functions applied to rotated features with $SO(2)$ linear layers for strict equivariance. The eSCN convolutions decompose $SO(3)$ tensor products during message passing into rotation and $SO(2)$ linear layers and therefore significantly reduce the complexity, enabling higher values of $L_{max}$ on large-scale datasets like OC20 (Chanussot* et al., 2021). EquiformerV2 (Liao et al., 2024b) adopts eSCN convolutions and proposes an improved version of Equiformer to better leverage the power of higher $L_{max}$. While higher $L_{max}$ can enhance the performance on large-scale datasets, the improvements are often evaluated using energy and forces from single-point DFT calculations instead of complex simulations. To translate the stronger performance to downstream simulation tasks, eSEN (Fu et al., 2025) incorporates design choices that can help achieve energy conservation during simulations. While effective, eSEN mainly follows the architecture of Equiformer(V1) but removes the attention. UMA (Wood et al., 2025) adopts eSEN-like models and explores how training across chemical domains can improve performance. In this work, we focus on architectural improvements enhancing efficiency, expressivity and generality and show that EquiformerV3 can consistently achieve better results on energy and force prediction and simulation-based tasks.

## B. Overall Architecture

**Equivariant Graph Attention.**   We illustrate equivariant graph attention with the proposed improvements in Figure 1(b). For each edge $(i, j)$, we first concatenate node embeddings $x_i$ and $x_j$ along the channel dimension and then rotate them with the permuted rotation matrices $\widetilde{D}_{ij}$ as described in Section 3.1. We then apply the first $SO(2)$ linear layer to obtain two sets of features: scalar features $f_{ij}^{(0)}$, containing only vectors of degree 0, and irreps features $f_{ij}^{(L)}$, containing components of all degrees up to $L_{max}$. To compute the attention weights $a_{ij}$, we apply layer normalization (Ba et al., 2016), a leaky ReLU layer and a linear layer to convert $f_{ij}^{(0)}$ into attention logits $z_{ij}$. Then, we incorporate smooth radius cutoff into attention

| Model | Counterexamples from Pozdnyakov et al. (2020) | | |
| --- | --- | --- | --- |
| | 2-body | 3-body (Fig. 1(b)) | 4-body (Fig. 2(f)) |
| Gate activation | | | |
|    Number of FFNs = 1 | 50.0 | 50.0 | 50.0 |
|    Number of FFNs = 2 | 50.0 | 50.0 | 50.0 |
|    Number of FFNs = 3 | 50.0 | 50.0 | 50.0 |
| $S^2$ activation | | | |
|    Number of FFNs = 1 | 50.0 | 50.0 | 50.0 |
|    Number of FFNs = 2 | 50.0 | 50.0 | 50.0 |
|    Number of FFNs = 3 | 50.0 | 50.0 | 50.0 |
| SwiGLU-$S^2$ activation | | | |
|    Number of FFNs = 1 | 100.0 | 50.0 | 50.0 |
|    Number of FFNs = 2 | 100.0 | 100.0 | 100.0 |
|    Number of FFNs = 3 | 100.0 | 100.0 | 100.0 |

*Table 5.* Results on the body order experiments in Joshi et al. (2023). The $k$-body counterexample contains two geometric graphs that are indistinguishable using $k$-body scalarization. "FFNs" denotes feedforward networks. We mark results where models successfully distinguish two geometric graphs in a counterexample and thus achieve 100% accuracy in green . Otherwise, they achieve 50% accuracy and are marked in red .

weights $a_{ij}$ as in Equation 4. For the value vectors $v_{ij}$, we apply the proposed SwiGLU-$S^2$ activation and the second $SO(2)$ linear layer. Finally, we combine $a_{ij}$, $v_{ij}$ and the envelope function envelope($||\vec{r}_{ij}||$) and rotate back to the original coordinate frame with $(\widetilde{D}_{ij})^{-1}$. Multiple attention heads are computed in parallel, implemented via a reshape operation.

**Feedforward Network.** As shown in Figure 1(d), we use the proposed SwiGLU-$S^2$ activation as the intermediate activation function. In addition, we apply two standard linear layers after ToSphere() and before FromSphere() for better mixing the information across channels of grid features.

**Embedding.** This module consists of atom embedding and edge-degree embeddings as illustrated in Figure 1(c). The atom embeddings maps the one-hot encoding of atom types into node features. For edge-degree embeddings, we apply a single $SO(2)$ linear layer followed by $(\widetilde{D}_{ij})^{-1}$ to incorporate information of edge vectors (or relative position vectors) and then perform sum aggregation to encode the information of node degrees.

**Output Head.** For graph-level predictions such as energy and stress, we apply feedforward networks with an intermediate gate activation (Weiler et al., 2018) to transform node irreps features, followed by aggregation over all the nodes. For node-wise predictions like forces and noise, we use equivariant graph attention.

## C. Body Order Experiment

We conduct the body order experiments in Joshi et al. (2023). The experiments evaluate models' ability to build distinguishing fingerprints for local neighborhoods and test whether GNNs with a single message passing can distinguish geometric graphs requiring higher body-order scalarization. Following the setting of local neighborhoods and a single message passing, we include only one attention block in models trained for the experiments. To incorporate higher body-order interactions, we stack one or multiple feedforward networks (FFNs) after the attention block, with each FFN containing one intermediate activation function. We train EquiformerV3 models with different activation functions and different numbers of FFNs to distinguish 2-body, 3-body and 4-body counterexamples from Pozdnyakov et al. (2020). The $k$-body counterexample contains two geometric graphs that are indistinguishable using $k$-body scalarization. The results are summarized in Table 5. Using the gate activation and $S^2$ activation in FFNs can only consider 2-body scalarization (i.e., the output scalars depend on only unordered set of pairwise distances) regardless of how many FFNs are included after the message passing and therefore cannot distinguish two local neighborhoods requiring body order greater than 2 (e.g., two local neighborhoods with identical pairwise distances (2-body) but different angles (3-body)). As the models cannot distinguish two geometric graphs in all the counterexamples, this results in an uniform accuracy of 50%. In contrast, using a single SwiGLU-$S^2$ activation can capture similar interactions to self tensor products (i.e., $x \otimes x$), achieving 3-body scalarization. Therefore, having one FFN with SwiGLU-$S^2$ activation can distinguish the 2-body counterexample, resulting in the accuracy of 100%. For higher body order, we can simply stack more FFNs. Note that stacking two consecutive FFNs is similar to $(x \otimes x) \otimes (x \otimes x) = x \otimes x \otimes x \otimes x$ and therefore includes 5-body scalarization.

| Activation function | $(R_\phi, R_\theta)$ | | | | | | |
|---|---|---|---|---|---|---|---|
| | (6, 6) | (8, 8) | (10, 10) | (12, 12) | (14, 14) | (16, 16) | (24, 24) |
| Gate activation | $1.43 \times 10^{-6}$ | $1.43 \times 10^{-6}$ | $1.43 \times 10^{-6}$ | $1.43 \times 10^{-6}$ | $1.43 \times 10^{-6}$ | $1.43 \times 10^{-6}$ | $1.43 \times 10^{-6}$ |
| $S^2$ activation | $9.95 \times 10^{-3}$ | $3.80 \times 10^{-4}$ | $5.43 \times 10^{-5}$ | $3.43 \times 10^{-6}$ | $7.30 \times 10^{-7}$ | $5.31 \times 10^{-7}$ | $5.31 \times 10^{-7}$ |
| SwiGLU-$S^2$ activation | $3.93 \times 10^{-2}$ | $1.05 \times 10^{-6}$ | $1.04 \times 10^{-6}$ | $1.04 \times 10^{-6}$ | $1.04 \times 10^{-6}$ | $1.04 \times 10^{-6}$ | $1.04 \times 10^{-6}$ |

*(a)* Equivariance errors when using $L_{max} = 2$.

| Activation function | $(R_\phi, R_\theta)$ | | | | | | |
|---|---|---|---|---|---|---|---|
| | (10, 10) | (12, 12) | (14, 14) | (16, 16) | (20, 20) | (24, 24) | (32, 32) |
| Gate activation | $1.31 \times 10^{-6}$ | $1.31 \times 10^{-6}$ | $1.31 \times 10^{-6}$ | $1.31 \times 10^{-6}$ | $1.31 \times 10^{-6}$ | $1.31 \times 10^{-6}$ | $1.31 \times 10^{-6}$ |
| $S^2$ activation | $1.07 \times 10^{-2}$ | $5.28 \times 10^{-3}$ | $3.96 \times 10^{-4}$ | $1.57 \times 10^{-4}$ | $1.35 \times 10^{-5}$ | $1.26 \times 10^{-6}$ | $1.17 \times 10^{-6}$ |
| SwiGLU-$S^2$ activation | $5.57 \times 10^{-2}$ | $2.72 \times 10^{-2}$ | $1.04 \times 10^{-6}$ | $1.04 \times 10^{-6}$ | $1.04 \times 10^{-6}$ | $1.04 \times 10^{-6}$ | $1.04 \times 10^{-6}$ |

*(b)* Equivariance errors when using $L_{max} = 4$.

| Activation function | $(R_\phi, R_\theta)$ | | | | | | |
|---|---|---|---|---|---|---|---|
| | (14, 14) | (16, 16) | (18, 18) | (20, 20) | (24, 24) | (28, 28) | (32, 32) |
| Gate activation | $1.31 \times 10^{-6}$ | $1.31 \times 10^{-6}$ | $1.31 \times 10^{-6}$ | $1.31 \times 10^{-6}$ | $1.31 \times 10^{-6}$ | $1.31 \times 10^{-6}$ | $1.31 \times 10^{-6}$ |
| $S^2$ activation | $1.05 \times 10^{-2}$ | $6.30 \times 10^{-3}$ | $3.40 \times 10^{-3}$ | $3.61 \times 10^{-4}$ | $9.76 \times 10^{-5}$ | $1.91 \times 10^{-5}$ | $2.04 \times 10^{-6}$ |
| SwiGLU-$S^2$ activation | $5.37 \times 10^{-2}$ | $3.28 \times 10^{-2}$ | $2.14 \times 10^{-2}$ | $1.24 \times 10^{-6}$ | $1.24 \times 10^{-6}$ | $1.24 \times 10^{-6}$ | $1.24 \times 10^{-6}$ |

*(c)* Equivariance errors when using $L_{max} = 6$.

*Table 6.* Equivariance errors when using different activations in feedforward networks. We consider maximum degree $L_{max} = 2, 4, 6$. For each $L_{max}$, we vary the numbers of grid points $(R_\phi, R_\theta)$ sampled along $\phi$ and $\theta$ and report the resulting equivariance errors. The equivariance errors of the gate activation are reported as the baseline of strict equivariance. We mark results achieving similar equivariance errors to the gate activation function in green . Otherwise, they are marked in red .

## D. Equivariance Errors of Different Activation Functions

We compare the equivariance errors of using different activation functions in feedforward networks (FFNs) and attention here. The equivariance errors in FFNs are summarized in Table 6. Compared to $S^2$ activation, the proposed SwiGLU-$S^2$ activation can achieve equivariance errors similar to the gate activation while reducing the number of grid points by more than $2\times$. For example, when $L_{max} = 2$, SwiGLU-$S^2$ activation requires $64 (= 8 \times 8)$ grid points while $S^2$ activation needs $144 (= 12 \times 12)$. Additionally, since the eSCN convolutions in attention discard components of irreps features with order $m > M_{max}$, where $M_{max}$ is a pre-defined hyper-parameters, we can further reduce the number of grid points in the SwiGLU-$S^2$ activation while maintaining similar equivariance errors. We compare the equivariance error in attention in Table 7. Particularly, we find that when using $L_{max} = 6$ and $M_{max} = 2$ as the default configuration on OC20, SwiGLU-$S^2$ activation requires $160 (= 8 \times 20)$ grid points in attention. Compared to EquiformerV2, which uses 324 grid points in attention, this reduces the complexity of sampling by $50.6\%$.

## E. Details of Experiments on OC20

We follow the definition of architectural hyper-parameters in EquiformerV2 (Liao et al., 2024b) to specify the base model (Index 7) in Table 1. The hyper-parameters are summarized in Table 8. The dimension of irreps features is denoted as $(L_{max}, C)$. All irreps features have degrees from 0 to $L_{max}$ and have $C$ channels for each degree. Different from EquiformerV2, here we use $d_{attn\_hidden}$ to specify the dimension after the SwiGLU-$S^2$ activation. Specifically, $f_{ij}^{(L)}$ originally has $2\times$ more channels and has dimension $(6, 128)$. After the SwiGLU-$S^2$ activation, the number of channels is halved, and therefore $f_{ij}^{(L)}$ has dimension $(6, 64)$. Similarly, $d_{ffn}$ specifies the hidden dimension in feedforward networks after the SwiGLU-$S^2$ activation. The training time and the number of parameters can be found in Table 1.

## F. Details of Experiments on OMat24

We follow the practice of direct pre-training and gradient fine-tuning by eSEN (Fu et al., 2025). During direct pre-training, we use regularizations like dropout (Srivastava et al., 2014) and stochastic depth (Huang et al., 2016) and the auxiliary task of denoising based on DeNS (Liao et al., 2024a). During gradient fine-tuning, we do not use any of the regularizations or

| Activation function | $(R_\phi, R_\theta)$ | | | | |
|---|---|---|---|---|---|
| | (14, 14) | (12, 14) | (10, 14) | (8, 14) | (6, 14) |
| Gate activation | $1.07 \times 10^{-6}$ | $1.07 \times 10^{-6}$ | $1.07 \times 10^{-6}$ | $1.07 \times 10^{-6}$ | $1.07 \times 10^{-6}$ |
| $S^2$ activation | $1.82 \times 10^{-5}$ | $1.82 \times 10^{-5}$ | $9.04 \times 10^{-5}$ | $3.12 \times 10^{-4}$ | $3.82 \times 10^{-2}$ |
| SwiGLU-$S^2$ activation | $1.43 \times 10^{-6}$ | $1.43 \times 10^{-6}$ | $1.43 \times 10^{-6}$ | $1.43 \times 10^{-6}$ | $6.72 \times 10^{-2}$ |

*(a)* Equivariance errors when using $L_{max} = 4$ and $M_{max} = 2$.

| Activation function | $(R_\phi, R_\theta)$ | | | | | |
|---|---|---|---|---|---|---|
| | (20, 20) | (16, 20) | (12, 20) | (10, 20) | (8, 20) | (6, 20) |
| Gate activation | $1.28 \times 10^{-6}$ | $1.28 \times 10^{-6}$ | $1.28 \times 10^{-6}$ | $1.28 \times 10^{-6}$ | $1.28 \times 10^{-6}$ | $1.28 \times 10^{-6}$ |
| $S^2$ activation | $2.06 \times 10^{-5}$ | $2.06 \times 10^{-5}$ | $2.03 \times 10^{-5}$ | $1.61 \times 10^{-4}$ | $5.87 \times 10^{-4}$ | $7.51 \times 10^{-2}$ |
| SwiGLU-$S^2$ activation | $1.74 \times 10^{-6}$ | $1.74 \times 10^{-6}$ | $1.74 \times 10^{-6}$ | $1.74 \times 10^{-6}$ | $1.74 \times 10^{-6}$ | $9.44 \times 10^{-2}$ |

*(b)* Equivariance errors when using $L_{max} = 6$ and $M_{max} = 2$.

*Table 7.* Equivariance errors when using different activations in attention. We consider maximum degree $L_{max} = 4, 6$ and maximum order $M_{max} = 2$. For each $L_{max}$, we start with the minimal number of grid points $(R_\phi, R_\theta)$ that enables the SwiGLU-$S^2$ activation to have comparable equivariance errors to the gate activation as in Table 6. Then, we further reduce $R_\phi$. We mark results achieving similar equivariance errors to the gate activation function in green . Otherwise, they are marked in red .

DeNS. We train two EquiformerV3 models with $L_{max} = 4$ and 6, and the hyper-parameters are in Table 9. We use 32 H100 GPUs for training. For $L_{max} = 4$, direct pre-training takes 2188 GPU-hours, and gradient fine-tuning takes 2594 GPU-hours. For $L_{max} = 6$, direct pre-training takes 3320 GPU-hours, and gradient fine-tuning takes 5877 GPU-hours.

## G. Details of Experiments on Matbench Discovery

**Training on Only MPtrj Dataset.**    We follow the practice of direct pre-training and gradient fine-tuning by eSEN (Fu et al., 2025). During direct pre-training, we use regularizations like dropout (Srivastava et al., 2014) and stochastic depth (Huang et al., 2016) and the auxiliary task of denoising based on DeNS (Liao et al., 2024a). During gradient fine-tuning, we do not use any of the regularizations or DeNS. The hyper-parameters are in Table 10. 16 H100 GPUs are used for training. Direct pre-training takes 599 GPU-hours, and gradient fine-tuning takes 471 GPU-hours.

**Fine-tuning OMat24-pretrained Models on MPtrj and Subsampled Alexandria Dataset.**    After pre-training EquiformerV3 with $L_{max} = 4$ on the OMat24 dataset, we fine-tune it with gradient methods on MPtrj and subsampled Alexandria datasets. We do not use any regularization or DeNS. Same as eSEN, one epoch of training consists of 8 copies of MPtrj dataset and 1 copy of subsampled Alexandria dataset. The hyper-parameters are in Table 11. 32 H100 GPUs are used for training, and the training time is 892 GPU-hours.

| Hyper-parameters | Value or description |
|---|---|
| Optimizer | AdamW |
| Learning rate scheduling | Cosine learning rate with linear warmup |
| Warmup epochs | 0.1 |
| Maximum learning rate | $2 \times 10^{-4}$ |
| Batch size | 64 |
| Number of epochs | 12 |
| Weight decay | $1 \times 10^{-3}$ |
| Dropout rate | 0.1 |
| Stochastic depth | 0.05 |
| Energy coefficient $\lambda_E$ | 4 |
| Force coefficient $\lambda_F$ | 100 |
| Gradient clipping norm threshold | 100 |
| Model EMA decay | 0.999 |
| Cutoff radius (Å) | 12 |
| Maximum number of neighbors | 20 |
| Radial basis function | Gaunssian |
| Number of radial bases | 128 |
| Dimension of hidden scalar features in radial functions $d_{edge}$ | $(0, 128)$ |
| Maximum degree $L_{max}$ | 6 |
| Maximum order $M_{max}$ | 2 |
| Number of Transformer blocks | 8 |
| Embedding dimension $d_{embed}$ | $(6, 128)$ |
| $f_{ij}^{(L)}$ dimension $d_{attn\_hidden}$ | $(6, 64)$ |
| Number of attention heads $h$ | 8 |
| $f_{ij}^{(0)}$ dimension $d_{attn\_alpha}$ | $(0, 64)$ |
| Value dimension $d_{attn\_value}$ | $(6, 16)$ |
| Hidden dimension in feedforward networks $d_{ffn}$ | $(6, 512)$ |
| Resolution of grid points in attention $(R_\phi, R_\theta)$ | $(8, 20)$ |
| Resolution of grid points in feedforward networks $(R_\phi, R_\theta)$ | $(20, 20)$ |

*Table 8.* Hyper-parameters for the base model setting on OC20 S2EF-2M dataset.

| Hyper-parameters | Direct pre-training | Gradient fine-tuning |
|---|---|---|
| Optimizer | AdamW | AdamW |
| Learning rate scheduling | Cosine learning rate with linear warmup | Cosine learning rate with linear warmup |
| Warmup epochs | 0.1 | 0.1 |
| Maximum learning rate | $2 \times 10^{-4}$ | $1 \times 10^{-4}$ |
| Batch size | 512 | 512 |
| Number of epochs | 4 | 2 |
| Weight decay | $1 \times 10^{-3}$ | $1 \times 10^{-3}$ |
| Dropout rate | 0.1 | 0.0 |
| Stochastic depth | 0.05 | 0.0 |
| Energy coefficient $\lambda_E$ | 20 | 20 |
| Force coefficient $\lambda_F$ | 20 | 20 |
| Stress coefficient $\lambda_s$ | 5 | 5 |
| Gradient clipping norm threshold | 100 | 100 |
| Model EMA decay | 0.999 | 0.999 |
| Cutoff radius (Å) | 6 | 6 |
| Maximum number of neighbors | 300 | 300 |
| Radial basis function | Gaunssian | Gaussian |
| Number of radial bases | 64 | 64 |
| Dimension of hidden scalar features in radial functions $d_{edge}$ | $(0, 128)$ | $(0, 128)$ |
| Maximum degree $L_{max}$ | $4, 6$ | $4, 6$ |
| Maximum order $M_{max}$ | 2 | 2 |
| Number of Transformer blocks | 7 | 7 |
| Embedding dimension $d_{embed}$ | $(4, 128), (6, 128)$ | $(4, 128), (6, 128)$ |
| $f_{ij}^{(L)}$ dimension $d_{attn\_hidden}$ | $(4, 32), (6, 32)$ | $(4, 32), (6, 32)$ |
| Number of attention heads $h$ | 8 | 8 |
| $f_{ij}^{(0)}$ dimension $d_{attn\_alpha}$ | $(0, 64)$ | $(0, 64)$ |
| Value dimension $d_{attn\_value}$ | $(4, 16), (6, 16)$ | $(4, 16), (6, 16)$ |
| Hidden dimension in feedforward networks $d_{ffn}$ | $(4, 512), (6, 512)$ | $(4, 512), (6, 512)$ |
| Resolution of grid points in attention $(R_\phi, R_\theta)$ | $(8, 14)$ for $L_{max} = 4$ $(8, 20)$ for $L_{max} = 6$ | $(8, 14)$ for $L_{max} = 4$ $(8, 20)$ for $L_{max} = 6$ |
| Resolution of grid points in feedforward networks $(R_\phi, R_\theta)$ | $(14, 14)$ for $L_{max} = 4$ $(20, 20)$ for $L_{max} = 6$ | $(14, 14)$ for $L_{max} = 4$ $(20, 20)$ for $L_{max} = 6$ |
| Probability of optimizing DeNS $p_{\text{DeNS}}$ | 0.5 | 0.0 |
| DeNS coefficient $\lambda_{\text{DeNS}}$ | 1 | 0.0 |
| Standard deviation of Gaussian noise $\sigma$ | 0.025 | 0.0 |
| Corruption ratio $r_{\text{DeNS}}$ | 0.5 | 0.0 |

*Table 9.* Hyper-parameters for the OMat24 dataset.

| Hyper-parameters | Direct pre-training | Gradient fine-tuning |
|---|---|---|
| Optimizer | AdamW | AdamW |
| Learning rate scheduling | Cosine learning rate with linear warmup | Cosine learning rate with linear warmup |
| Warmup epochs | 0.1 | 0.1 |
| Maximum learning rate | $2 \times 10^{-4}$ | $5 \times 10^{-5}$ |
| Batch size | 512 | 512 |
| Number of epochs | 70 | 10 |
| Weight decay | $1 \times 10^{-3}$ | $1 \times 10^{-3}$ |
| Dropout rate | 0.1 | 0.0 |
| Stochastic depth | 0.05 | 0.0 |
| Energy coefficient $\lambda_E$ | 20 | 20 |
| Force coefficient $\lambda_F$ | 20 | 20 |
| Stress coefficient $\lambda_s$ | 5 | 5 |
| Gradient clipping norm threshold | 100 | 100 |
| Model EMA decay | 0.999 | 0.999 |
| Cutoff radius (Å) | 6 | 6 |
| Maximum number of neighbors | 300 | 300 |
| Radial basis function | Gaunssian | Gaussian |
| Number of radial bases | 10 | 10 |
| Dimension of hidden scalar features in radial functions $d_{edge}$ | $(0, 128)$ | $(0, 128)$ |
| Maximum degree $L_{max}$ | 4 | 4 |
| Maximum order $M_{max}$ | 2 | 2 |
| Number of Transformer blocks | 7 | 7 |
| Embedding dimension $d_{embed}$ | $(4, 128)$ | $(4, 128)$ |
| $f_{ij}^{(L)}$ dimension $d_{attn\_hidden}$ | $(4, 32)$ | $(4, 32)$ |
| Number of attention heads $h$ | 8 | 8 |
| $f_{ij}^{(0)}$ dimension $d_{attn\_alpha}$ | $(0, 64)$ | $(0, 64)$ |
| Value dimension $d_{attn\_value}$ | $(4, 16)$ | $(4, 16)$ |
| Hidden dimension in feedforward networks $d_{ffn}$ | $(4, 512)$ | $(4, 512)$ |
| Resolution of grid points in attention $(R_\phi, R_\theta)$ | $(8, 14)$ | $(8, 14)$ |
| Resolution of grid points in feedforward networks $(R_\phi, R_\theta)$ | $(14, 14)$ | $(14, 14)$ |
| Probability of optimizing DeNS $p_{\text{DeNS}}$ | 0.5 | 0.0 |
| DeNS coefficient $\lambda_{\text{DeNS}}$ | 10 | 0.0 |
| Standard deviation of Gaussian noise $\sigma$ | 0.025 | 0.0 |
| Corruption ratio $r_{\text{DeNS}}$ | 0.5 | 0.0 |

*Table 10.* Hyper-parameters for the MPtrj dataset.

| Hyper-parameters | Gradient fine-tuning |
| --- | --- |
| Optimizer | AdamW |
| Learning rate scheduling | Cosine learning rate with linear warmup |
| Warmup epochs | 0.1 |
| Maximum learning rate | $5 \times 10^{-5}$ |
| Batch size | 256 |
| Number of epochs | 2 |
| Weight decay | $1 \times 10^{-3}$ |
| Dropout rate | 0.0 |
| Stochastic depth | 0.0 |
| Energy coefficient $\lambda_E$ | 20 |
| Force coefficient $\lambda_F$ | 20 |
| Stress coefficient $\lambda_s$ | 5 |
| Gradient clipping norm threshold | 100 |
| Model EMA decay | 0.999 |
| Cutoff radius (Å) | 6 |
| Maximum number of neighbors | 300 |
| Radial basis function | Gaussian |
| Number of radial bases | 64 |
| Dimension of hidden scalar features in radial functions $d_{edge}$ | $(0, 128)$ |
| Maximum degree $L_{max}$ | 4 |
| Maximum order $M_{max}$ | 2 |
| Number of Transformer blocks | 7 |
| Embedding dimension $d_{embed}$ | $(4, 128)$ |
| $f_{ij}^{(L)}$ dimension $d_{attn\_hidden}$ | $(4, 32)$ |
| Number of attention heads $h$ | 8 |
| $f_{ij}^{(0)}$ dimension $d_{attn\_alpha}$ | $(0, 64)$ |
| Value dimension $d_{attn\_value}$ | $(4, 16)$ |
| Hidden dimension in feedforward networks $d_{ffn}$ | $(4, 512)$ |
| Resolution of grid points in attention $(R_\phi, R_\theta)$ | $(8, 14)$ |
| Resolution of grid points in feedforward networks $(R_\phi, R_\theta)$ | $(14, 14)$ |
| Probability of optimizing DeNS $p_{\text{DeNS}}$ | 0.0 |
| DeNS coefficient $\lambda_{\text{DeNS}}$ | 0.0 |
| Standard deviation of Gaussian noise $\sigma$ | 0.0 |
| Corruption ratio $r_{\text{DeNS}}$ | 0.0 |

*Table 11.* Hyper-parameters for fine-tuning OMat24-pretrained models on MPtrj and subsampled Alexandria datasets.

