# OpenReview forum: "EquiformerV3: Scaling Efficient, Expressive and General SE(3)-Equivariant Graph Attention Transformers"
_ICML.cc/2026/Conference — Submitted to ICML 2026_

### Official Review · Reviewer_b7nK · 2026-03-07

**Soundness:** 3
**Presentation:** 3
**Significance:** 3
**Originality:** 2
**Overall Recommendation:** 4
**Confidence:** 3

**Summary:**

The paper introduces a further development of the Equiformer family. The main contributions are: 1) Code level optimizations  using better compilation and fusion of operations. 2) Hyperparameter tuning and slight variations of the normalization operation 3) A smoother attention mechanism for better derivative properties 4) Proposal of the SwiGLU-S^2 activation function 5) Extensive testing on different datasets
The model achieves superior accuracy and much faster training compared to competing methods

**Compliance With Llm Reviewing Policy:**

Affirmed.

**Final Justification:**

While I think the technical contributions are mostly incremental, they have been combined in a skillful way that does lead to a very strongly performing model. I therefore increased my score and would recommend the work to be accepted.

**Key Questions For Authors:**

- Table 1: Why is parameter count so much higher with SwiGLU-S^2
- I am not sure, but it seems like SwigGLU-S^2 is identical to a tensorproduct followed by gating, which is a known operation. In particular:
$$
SwiGLU-S^2(x_{scalar}, x_1, x_2) = \text{ FromSphere}(\text{Sigmoid}(x_{scalar}) * x_1 * x_2) =
$$
$$
\int \int \text{Sigmoid}(x_{scalar}) * x_1 * x_2 = \text{Sigmoid}(x_{scalar})  \int \int x_1 * x_2 = \text{Sigmoid}(x_{scalar}) x_1 \otimes x_2
$$
The proposed formulation $\int \int \text{Sigmoid}(x_{scalar}) * x_1 * x_2$ even seems slower than the naive and known pattern $\text{Sigmoid}(x_{scalar}) (x_1 \otimes x_2)$, as the the integral formulation takes $R_\theta R_\phi C$ multiplications and the naive one takes only $(L+1)^2 C$ multiplications which is generally less.

**Limitations:**

-

**Strengths And Weaknesses:**

Strengths:
- The paper improves on an already competitive model in an important field
- In particular, the training time reduction and error reduction in derivative properties seem strong
- The merged layer norm and SwiGLU-S^2 activation seem to lead to significantly better properties compared to EquiformerV2
- The model is extensively tested on large datasets


Weaknesses:
- The majority of improvements leading to faster training stem from better engineering and hyperparameters, which are valuable contributions but not very original.
- One of the major contributions, the SwiGLU-S^2 seems to be identical to previously known operations, just written more complicated and theoretically less efficient. I might be misunderstanding something, but an explanation of how it is different would be good. Please see the questions section below
- Inference timings are critical in MD and should be reported, not just train timings.
- Smooth attention and SwiGLU-S^2 are hypothesized to be the reason for better derivative properties like $\kappa$. However the ablation study was only done for forces and energies, but the advantage there is relatively minor. It would be great to have an ablation on $\kappa$ to see which of the proposed changes is most responsible for the improvement

---

> ### Author Rebuttal · Authors · 2026-03-31
>
> > 1. [Weakness 2 and Question 2] The SwiGLU-$S^2$ seems to be identical to known operations but slower.
>
> The proposed SwiGLU-$S^2$ activation is equivalent to first applying tensor products and then applying gating.
>
> Our proposed formula is **faster** in terms of theoretical complexity (Line 263 – 268, right column) and in current implementations and provides a unifying perspective that **explains why tensor products with gating are strictly more expressive than gate or $S^2$ activations** (Line 282 – 299, left column and Table 5).
>
> The standard, naive tensor product (i.e., $z = x \otimes y$) has compute complexity $\mathcal{O}(L_{max}^6)$ [1], not $\mathcal{O}((L_{max} + 1) ^2) = \mathcal{O}(L_{max} ^2)$ as mentioned by the reviewer, given that $x, y \in \mathbb{R}^{(L_{max} + 1)^2}$. Our formula using fast tensor products [2, 3, 4] has complexity $\mathcal{O}(R_{\phi} R_{\theta} (L_{max} + 1) ^2) = \mathcal{O}(L_{max} ^4)$ as $R_{\phi}$ and $R_{\theta}$ have linear dependence on $L_{max}$. We note that **[2, 3, 4] also provide similar analysis on the complexity and show that fast tensor products are more efficient in practice**.
>
> We compare the inference speed of standard tensor products implemented in  **e3nn** [5] and **fast tensor products (ours)** here.
>
> $L_{max}$ = 4
> | | Time (ms) | Memory (MB) |
> |--|--|--|
> | e3nn | 15.49 | 432.2 |
> | **fast tensor product (ours)** | **2.98** | **108.2** |
>
> $L_{max}$ = 6
> | | Time (ms) | Memory (MB) |
> |--|--|--|
> | e3nn | 71.33 | 1608.4 |
> | **fast tensor product (ours)** | **6.48** | **204.6** |
>
> **Our fast tensor product formula is faster and more memory-efficient.**
>
> [1] Passaro et al. Reducing SO(3) Convolutions to SO(2) for Efficient Equivariant GNNs. ICML 2023.
>
> [2] Xin et al. Fast and Accurate Spherical Harmonics Products. ACM Transactions on Graphics 2021.
>
> [3] Luo et al. Enabling Efficient Equivariant Operations in the Fourier Basis via Gaunt
> Tensor Products. ICLR 2024.
>
> [4] Xie et al. The Price of Freedom: Exploring Expressivity and Runtime Tradeoffs in Equivariant Tensor Products. ICML 2025.
>
> [5] https://github.com/e3nn/e3nn
>
>
> ---
>
> > 2. [Weakness 4] Smooth attention and SwiGLU-$S^2$ are hypothesized to be the reason for better derivative properties like $\kappa$. An ablation on $\kappa$.
>
> The **ablation study** on how smooth radius cutoff and strict equivariance affect energy conservation and downstream properties like $\kappa$ was **already conducted by eSEN** [1] (Figure 4 and Figure 6 in [1]). This work follows their findings to design EquiformerV3. Our model achieves better results on $\kappa$ and MDR Phonon benchmark (Please see our **response 2. to Reviewer bPMq**), which empirically confirms what [1] identifies. We will make this point more clear.
>
> [1] Fu et al. Learning Smooth and Expressive Interatomic Potentials for Physical Property Prediction. ICML 2025.
>
> ---
>
> > 3. [Weakness 3] Inference time.
>
> We add the **inference time on OC20** to Table 1 as below.
> |  | Energy MAE | Force MAE | Inference time per sample (ms) | Training time (GPU-hours) |
> |---|---|---|---|---|
> | Index 2 (EquiformerV2) | 242 | 19.73 | 19.20 | 270 |
> | Index 3 | 242 | 19.73 | 17.33 | 154 |
> | Index 4 | 236 | 19.28 | 17.33 | 150 |
> | Index 5 | 209 | 18.96 | 17.87 | 163 |
> | Index 6 | 213 | 18.82 | 17.87 | 163 |
> | Index 7 (EquiformerV3) | 201 | 18.15 | 20.05 | 171 |
>
> EquiformerV3 has similar inference speed to EquiformerV2 (20.05 vs 19.20 ms) despite using 37% more parameters and achieving significantly better accuracy. The 1.75× training speedup from `torch.compile()` (Index 2 versus 3) does not translate to a comparable inference speedup, as compilation behaves differently during training and inference. We leave inference-specific optimization such as custom CUDA kernels to future work.
>
> Besides, we follow eSEN’s setting to **compare the inference speed and peak memory of eSEN and EquiformerV3 models in Table 4**.
> |  | Time per iteration (ms) | Peak memory (MB) | Number of parameters |
> |---|---|---|---|
> | eSEN | 461 | 9259 | 30M |
> | EquiformerV3  | **318** | **8604** | 30M |
>
> **EquiformerV3 improves upon eSEN (ICML 2025 Oral) on all reported metrics including Matbench Discovery, MDR Phonon, inference speed, and memory.**
>
>
> ---
>
> > 4. [Weakness 1] Novelty.
>
> Please see our **response 3. to Reviewer bPMq**.
>
>
> ---
>
> > 5. [Question 1] Table 1: Why is parameter count so much higher with SwiGLU-S^2?
>
> Since the SwiGLU-$S^2$ activation halves the number of channels, we increase the number of input channels by 2× so that subsequent layers remain the same (Line 343 – 348, left column).
>
> Using SwiGLU-S² activation under similar numbers of parameters maintains similar performance gain and is faster:
> | Model | Energy | Force | Number of parameters | Training time (GPU-hours) |
> |--|--|--|--|--|
> | $S^2$ (Index 6 in Table 1) | 213 | 18.82 | 66M | 163 |
> | SwiGLU-$S^2$ (Index 7 in Table 1) | 201 | 18.15 | 91M | 171 |
> | SwiGLU-$S^2$ (Similar number of parameters to $S^2$) | 203 | 18.19 | 68M | 155 |

---

> > ### Author Rebuttal · Reviewer_b7nK · 2026-04-01
> >
> > > Our fast tensor product formula is faster and more memory-efficient.
> >
> > I might have been unclear with my question: I don't doubt that the integral formulation of the tensorproduct is faster, this has been clearly demonstrated by previous work. My impression was that one of the contributions of this work is to define SwiGLU-S^2 as:
> >
> > $$ \text{FromSphere}(\text{Sigmoid}(x_{scalar}) x_1 * x_2) \quad (A)$$
> >
> > instead of the already known pattern
> >
> > $$ \text{Sigmoid}(x_{scalar})  \text{FromSphere}(x_1 * x_2)   \quad (B)$$
> >
> > However, I dont see why the combined operation (A) as proposed is beneficial over just using a tensorproduct with the fast integral formula, followed by gating (B). My comment about the speed was referring to (A) requiring $R_\theta R_\phi C$ multiplications to apply the gating, while (B) requires $(L+1)^2 C$ multiplications to apply the gating. With usual parameters we have $R_\theta R_\phi C > (L+1)^2 C$, making the proposed formulation more expensive in theory. While I dont expect a large performance difference with either (A) or (B), I struggle to see what the contribution is.
> >
> > > [Weakness 1] Novelty. Please see our response 3. to Reviewer bPMq. ... The ablation study on how smooth radius cutoff and strict equivariance affect energy conservation and downstream properties like  was already conducted by eSEN [1] (Figure 4 and Figure 6 in [1]). This work follows their findings to design EquiformerV3.
> >
> > If the work mainly combines techniques from previous models it becomes unclear where the advantage over these previous models comes from. What transferable insights can we learn that makes EquiformerV3 better than previous models? This is important to understand if the improvements come mainly from more invested compute in either model training or hyperparameter tuning, or if the architecture itself is superior.

---

> > > ### Author Response · Authors · 2026-04-03
> > >
> > > > 6. [Acknowledgement 1] Clarification on which formula is faster in the original review.
> > >
> > > In [Question 1], the reviewer compared $\int \int \text{Sigmoid}(x_{scalar}) x_1^{grid} x_2^{grid}$ and $\text{Sigmoid}(x_{scalar}) x_1 \otimes x_2$ instead of what is described in the rebuttal acknowledgement.
> > > As in our response 1., the former has complexity $\mathcal{O}(L_{max}^4)$, and the latter has complexity $\mathcal{O}(L_{max}^6)$ because of $\otimes$.
> > >
> > > ---
> > >
> > > > 7. [Acknowledgement 1] Formula (A) and (B).
> > >
> > > Formula (A) ($\text{FromSphere}(\text{Sigmoid}(x_{scalar}) \cdot x_1^{grid} \cdot x_2^{grid})$) and formula (B) ($\text{Sigmoid}(x_{scalar}) \cdot \text{FromSphere}(x_1^{grid} \cdot x_2^{grid}$)) are the same mathematically, and formula (B) is slightly better in terms of complexity.
> > >
> > > However, **formula (A) provides the analysis on why the proposed SwiGLU-$S^2$ activation is better than the activations used in previous equivariant networks (Line 280 – 299, left column).** We reiterate the analysis below and compare the following activation:
> > >
> > > (a) (proposed) SwiGLU-$S^2$:= $\text{Sigmoid}(x_{scalar}) \cdot x_1^{grid} \cdot x_2^{grid}$
> > >
> > > (b) $S^2$ activation with SwiGLU:= $\text{SiLU}(x_1^{grid}) \cdot x_2^{grid}$ = $\text{Sigmoid}(x_1^{grid}) \cdot x_1^{grid} \cdot x_2^{grid}$
> > >
> > > (c) $S^2$ activation with SiLU:= $\text{SiLU}(x^{grid})$
> > >
> > > (d) Gate activation
> > >
> > > The analysis is:
> > >
> > > (i) (a) is comparable to (b) as using $x_{scalar}$ or $x_1^{grid}$ as nonlinearity gives similar performance.
> > >
> > > (ii) (b) is better than (c) since SwiGLU is better than SiLU as shown in many works on LLMs.
> > >
> > > (iii) (c) is better than (d) as shown in EquiformerV2.
> > >
> > > (iv) Thus, (a) ~ (b) > (c) > (d).
> > >
> > > **Formula (A) shows why the proposed SwiGLU-$S^2$ activation is better than (c) and (d) used in previous models.** In contrast, **it can be difficult to tell if formula (B) (i.e., tensor products with gating) can be better than $S^2$ activation.**
> > >
> > > In our implementation, we use formula (B) in attention, and we see little difference, same as the reviewer mentioned. However, in the feedforward networks (FFNs), we use formula (A) as we apply standard linear operations to grid features before and after the activation. Specifically, given node irreps feature $x$ and scalar feature $x_{scalar}$, the FFNs compute the output $z$ as:
> > >
> > > $x^{grid} \leftarrow \text{ToSphere}(x)$
> > >
> > > $y^{grid} \leftarrow Linear_1(x^{grid}) \cdot Linear_2(x^{grid})$
> > >
> > > $y^{grid} \leftarrow y^{grid} \cdot \text{Sigmoid}(x_{scalar})$
> > >
> > > $z^{grid} \leftarrow Linear_3(y^{grid})$
> > >
> > > $z \leftarrow \text{FromSphere}(z^{grid})$
> > >
> > > In short, (A) and (B) are minor implementation differences that would not affect how effective the proposed architecture is.
> > >
> > > ---
> > >
> > > > 8. [Acknowledgement 1] “I struggle to see what the contribution is.”
> > >
> > > We provide:
> > >
> > > (1) The analysis on why SwiGLU-$S^2$ is better than all the activation functions used by previous works as reiterated in the above response.
> > >
> > > (2) The theoretical justification (Section 3 and Table 5)
> > >
> > > (3) Empirical results (Table 1)
> > >
> > > These shows why SwiGLU-$S^2$ activation is theoretically and empirically a better architectural choice than the activations used by all previous models.
> > >
> > > As mentioned by the reviewer in the strengths of the original review, “the merged layer norm and SwiGLU-S^2 activation seem to lead to significantly better properties compared to EquiformerV2”. We believe the analysis, theory, and empirical results above explain why this is the case.
> > >
> > > ---
> > >
> > > > 9. [Acknowledgement 2] Unclear where the advantage comes from.
> > >
> > > Table 1 shows how each architectural improvement enhances results. Index 2 – Index 7 only differ in architectures.
> > >
> > > ---
> > >
> > > > 10. [Acknowledgement 2] transferable insights that makes EquiformerV3 better
> > >
> > > - Attention with smooth cutoff: **Previous models do not use attention**, one of the most important operations in deep learning, for MD simulations and learning smooth PES **since they cannot ensure its smooth cutoff**. This work addresses this issue and shows the overall effectiveness of attention.
> > >
> > > - SwiGLU-$S^2$ activation: Previous equivariant networks use strictly equivariant gate activation for MD simulations and learning smooth PES. While $S^2$ activation is stronger, it is approximately equivariant and therefore can prevent learning smooth PES. In contrast, the proposed SwiGLU-$S^2$ addresses the limitations of both activations – SwiGLU-$S^2$ activation is strictly equivariant and the strongest.
> > >
> > > ---
> > >
> > > > 11. [Acknowledgement 2] improvements from more compute in model training or hyperparameter tuning
> > >
> > > **Our model is more accurate and faster as mentioned in the summary and strength of the original review.**
> > >
> > > Comparing Index 2 and Index 7 in Table 1, our model has better results while taking 1.6x less training time and using the same hyper-parameters.
> > >
> > > Compared with eSEN and UMA-M in Table 4, our model is more accurate and has faster inference speed than both. Moreover, our model takes 22x less training time than UMA-M.

---

### Official Review · Reviewer_aXYy · 2026-03-10

**Soundness:** 2
**Presentation:** 3
**Significance:** 3
**Originality:** 2
**Overall Recommendation:** 3
**Confidence:** 3

**Summary:**

This paper presents EquiformerV3, an SE(3)-equivariant graph attention transformer for atomistic modeling that builds on the EquiformerV2 line of work. The paper aims to improve the prior model along three main directions: efficiency, expressivity, and generality. Concretely, it introduces implementation-level optimizations to reduce training cost, proposes several architectural refinements such as merged equivariant layer normalization and a smooth cutoff attention mechanism, and introduces a new SwiGLU-$S^2$ activation intended to improve many-body expressivity while reducing the $S^2$ grid resolution needed to maintain low equivariance error. The paper evaluates these modifications on OC20, OMat24, and Matbench Discovery, and reports improved accuracy-efficiency trade-offs as well as strong results on metrics that are intended to reflect smoother potential energy surfaces and better higher-order derivative behavior.

**Compliance With Llm Reviewing Policy:**

Affirmed.

**Final Justification:**

The rebuttal addressed several of my concerns, including the parameter-matched SwiGLU-S2 comparison and the ablation under consistent energy targets. Two concerns remain.

First, kSRME and MDR Phonon reflect the full model, making it difficult to attribute gains specifically to smooth cutoff attention.

Second, several key design choices are explicitly inherited from prior work, and some headline claims read somewhat broader than the in-paper evidence directly supports.

My overall recommendation remains unchanged, and I appreciate the authors' thorough engagement throughout the discussion.

**Key Questions For Authors:**

1. A parameter-matched comparison for the final SwiGLU-$S^2$ model would be very helpful. Since the final comparison changes both the activation and overall model capacity, such a control would clarify how much of the gain comes from the proposed activation itself. A clear result here would strengthen my assessment of the paper’s soundness.
2. More direct evidence for the benefit of smooth cutoff attention on PES smoothness or higher-order derivative behavior would strengthen the paper. For example, results on force constants, NVE energy drift, or a direct continuity test near the cutoff would make this motivation much more convincing and could positively affect my evaluation.
3. An end-to-end grid-resolution ablation for both S$^2$ and SwiGLU-$S^2$ under otherwise matched settings would be valuable. The appendix analysis of equivariance error is helpful, but downstream performance and runtime comparisons would better clarify whether reduced grid cost is a robust advantage of the proposed design.
4. A brief clarification on how to interpret the OC20 ablation after the target-formulation change would be appreciated. Because this change appears relatively early in the ablation path and seems to contribute a noticeable improvement, it would help to better understand how the later architectural gains should be interpreted.

**Limitations:**

yes

**Strengths And Weaknesses:**

### Strengths
* The paper addresses an important and timely problem. Improving SE(3)-equivariant atomistic transformers is highly relevant for large-scale interatomic potential learning, and the paper studies this question on strong and widely used benchmarks, including OC20, OMat24, and Matbench Discovery.
* The empirical evaluation is broad, and the OC20 ablation is useful. One strong aspect of the paper is that it decomposes the progression from EquiformerV2 to EquiformerV3 into several concrete changes. This makes the contribution easier to interpret than in many papers where multiple modifications are introduced at once with limited ablation.
* Some of the design choices are thoughtful and potentially useful beyond this specific model. In particular, incorporating the smooth cutoff directly inside the attention softmax is a technically interesting idea, since it directly addresses discontinuities induced by normalization over neighbors. Similarly, the proposed SwiGLU-$S^2$ activation appears to be motivated by a meaningful attempt to balance expressivity and equivariance preservation.

### Weaknesses
* While the ablation study is helpful and appreciated, some of the improvements are slightly difficult to attribute cleanly to specific architectural modifications. In the OC20 ablation sequence, the prediction target is changed early in the progression (from adsorption energy to total energy), which the authors note can improve accuracy. Because this change appears together with the architectural modifications in the same progression, it becomes somewhat harder to isolate the incremental effect of the subsequent architectural design choices. A brief clarification or an additional comparison under a consistent target formulation would make the architectural contribution easier to interpret.
* The smooth cutoff attention mechanism is well motivated, but its most important claimed benefit would benefit from more direct validation. The paper argues that this design is important for smooth PES modeling and higher-order derivative-related behavior. However, in the current version, the supporting evidence is somewhat indirect. The effect on standard single-point metrics appears modest, and I would have found the claim substantially stronger with a more targeted experiment on, for example, energy conservation, force constants, or a direct continuity test near the cutoff.
* The originality is moderate overall, and a few claims could be phrased more carefully. My overall impression is that this paper is a strong and thoughtful extension of EquiformerV2 rather than a fundamentally new modeling family. That is still valuable. However, because many ingredients are inherited from or closely related to prior work, I believe the paper would benefit from slightly more careful phrasing in places where the claims may otherwise read as somewhat stronger than the current evidence fully establishes.
* The final SwiGLU-$S^2$ comparison would also be easier to interpret with a parameter-matched control, since the current comparison changes the activation together with model capacity and grid configuration.

---

> ### Author Rebuttal · Authors · 2026-03-31
>
> > 1. [Weakness 2 and Question 2] Experiment on energy conservation, force constants, or a direct continuity test near the cutoff.
>
> Please see our **response 1. and 2. to Reviewer bPMq** for more details.
>
> For force constants, that is exactly what the thermal conductivity task [2] in Matbench Discovery is measuring (Line 410 – 420, left column). We achieve the best results on the corresponding $\kappa_{\text{SRME}}$ metric (Table 3 and Table 4).
>
> For direct continuity test, Equation (4) mathematically shows the smooth transition when atoms enter or leave a radius cutoff. We achieve energy conservation by construction and demonstrate the best results on $\kappa_{\text{SRME}}$ and MDR Phonon benchmark, which empirically confirms the smoothness of learned PES.
>
> [1] Fu et al. Learning Smooth and Expressive Interatomic Potentials for Physical Property Prediction. ICML 2025.
>
> [2] Póta et al. Thermal Conductivity Predictions with Foundation Atomistic Models. ArXiv 2024.
>
>
> ---
>
> > 2. [Weakness 1 and Question 4] Clarification on the performance gain on OC20 2M dataset.
>
> As we mentioned in the paper (Line 307 – 316, right column), we switch to total energy targets because predicting total energy prevents the ill-posed issues caused by adsorption energies, which enables better improvements on energy MAE.
>
> For the ablation study on the proposed architectures (Index 3 – 7 in Table 1), we use the same total energy targets to isolate the effect of different energy targets.
>
>
> ---
>
> > 3. [Weakness 1 and Question 4] Comparison under a consistent target formulation.
>
> We compare EquiformerV2 and EquiformerV3 under the setting of total and adsorption energy prediction below.
>
> Total energy:
> |  | Energy | Force |
> |---|---|---|
> | EquiformerV2 (Index 2 in Table 1) | 242 | 19.73 |
> | EquiformerV3 (Index 7 in Table 1) | 201 | 18.15 |
>
> Adsorption energy:
> |  | Energy | Force |
> |---|---|---|
> | EquiformerV2 (Index 1 in Table 1) | 296 | 21.23 |
> | EquiformerV3  | 274 | 18.91 |
>
> The proposed architectural improvements achieve consistent performance gain under both energy targets.
>
>
> ---
>
> > 4. [Weakness 2] The paper argues that smooth cutoff is important for smooth PES modeling and higher-order derivatives.
>
> The argument is from eSEN [1], and they conducted a thorough ablation study on how smooth radius cutoff and strict equivariance affect energy conservation, the smoothness of PES, and downstream properties like $\kappa$ (Figure 4 and Figure 6 in [1]). This work mainly follows their findings to design EquiformerV3. Our model achieves clearly better results on $\kappa$ and MDR Phonon benchmark (Please see our **response 2. to Reviewer bPMq**), which test the smoothness of PES and higher-order derivatives. We will make this point more clear.
>
> [1] Fu et al. Learning Smooth and Expressive Interatomic Potentials for Physical Property Prediction. ICML 2025.
>
>
> ---
>
> > 5. [Weakness 4 and Question 1] SwiGLU-$S^2$ comparison under a parameter-matched control.
>
> We reduce the hidden size in attention from 64 to 40 (Row 3 in the table below) so that the SwiGLU-$S^2$ variant has roughly the same number of parameters as the $S^2$ variant. The results are:
> | Model | Energy | Force | Number of parameters | Training time (GPU-hours) |
> |--|--|--|--|--|
> | $S^2$ (Index 6 in Table 1) | 213 | 18.82 | 66M | 163 |
> | SwiGLU-$S^2$ (Index 7 in Table 1) | 201 | 18.15 | 91M | 171 |
> | SwiGLU-$S^2$ (Similar number of parameters to $S^2$) | 203 | 18.19 | 68M | 155 |
>
> Using SwiGLU-$S^2$ activation under similar numbers of parameters maintains similar performance gain and is faster.
>
>
> ---
>
> > 6. [Question 3] An end-to-end grid-resolution ablation for both $S^2$ and SwiGLU-$S^2$ under matched settings.
>
> We increase the grid resolution of $S^2$ activation (Row 2) for strict equivariance, and the comparison on OC20 is below.
> |  | Grid resolution $R_{\phi}, R_{\theta}$ | Strict Equivariance | Energy MAE | Force MAE | Training time (GPU-hours) | Training memory usage | Number of parameters |
> |---|---|---|---|---|---|---|---|
> | $S^2$ activation (Index 6 in Table 1) | (18, 18) |  | 213 | 18.82 | 163 | 28GB | 66M |
> | $S^2$ activation (higher grid resolution) | (32, 32) | $\checkmark$ | 215 | 18.85 | 276 | 63GB | 66M |
> | SwiGLU-$S^2$ activation (Index 7 in Table 1) | See Table 8 | $\checkmark$ | 201 | 18.15 | 171 | 28GB | 91M |
>
> (1) Since the grid resolution is already high, using even higher resolution for strict equivariance gives no gain to $S^2$ activation.
>
> (2) **When strict equivariance is required, which is necessary to running MD simulations or learning smooth PES (e.g., Table 3 and Table 4 and MDR Phonon benchmark), SwiGLU-$S^2$ activation is strictly better – more accurate, faster and more memory-efficient.**
>
>
> ---
>
> > 7. [Weakness 3] The paper would benefit from slightly more careful phrasing.
>
> We are happy to revise specific phrasing if the reviewer can point to particular claims they feel are overstated.

---

> > ### Author Rebuttal · Reviewer_aXYy · 2026-04-02
> >
> > The rebuttal was helpful and clarified several of my concerns. I appreciate the additional explanations and comparisons. My overall assessment remains unchanged.

---

> > > ### Author Response · Authors · 2026-04-03
> > >
> > > We thank the reviewer for the efforts and want to follow up.
> > >
> > > In the review, you noted that:
> > >
> > > - Q2 (**primary concern**): … force constants … **“could positively affect my evaluation.”**
> > >
> > > - Q1: A parameter-matched comparison… **“A clear result here would strengthen my assessment of the paper’s soundness.”**
> > >
> > > We believe **we address both in the rebuttal**.
> > >
> > > We respectfully note that:
> > > - Q2 (**primary concern**): **The evaluation on force constants was already present in the submitted paper.** The $\kappa_{\text{SRME}}$ in Tables 3 and Table 4 is computed directly from second- and third-order force constants (Line 415–420). We additionally provide MDR Phonon results (SOTA on all four metrics) and MD energy conservation experiments.
> > > - Q1: We provide a parameter-matched comparison in the rebuttal and show that the SwiGLU-$S^2$ maintains similar performance gain.
> > > - Q3: Under strict equivariance, our SwiGLU-$S^2$ activation is strictly better in accuracy, speed, and memory.
> > > - Q4: EquiformerV3 shows consistent performance gain under both energy targets.
> > >
> > > Moreover, the **review summary** acknowledges “strong results on metrics that are intended to reflect smoother potential energy surfaces and better higher-order derivative behavior”, which is precisely the **evidence requested in the primary concern Q2**.
> > >
> > > We would welcome any remaining concerns, as “Partially resolved - I have follow-up questions” was selected but no follow-up questions were provided.

---

### Official Review · Reviewer_cq5P · 2026-03-12

**Soundness:** 2
**Presentation:** 3
**Significance:** 2
**Originality:** 2
**Overall Recommendation:** 3
**Confidence:** 5

**Summary:**

This paper presents EquiformerV3, a new SE(3)-equivariant graph attention Transformer for 3D atomistic modeling. The method combines several changes over EquiformerV2, including software optimization, Equivariant Merged Layer Normalization, smooth radius cutoff attention, stronger FFNs, and the proposed SwiGLU-S² activation. The reported results on OC20, OMat24, and Matbench Discovery are strong.

**Compliance With Llm Reviewing Policy:**

Affirmed.

**Final Justification:**

I still remain two main Concerns.
- The evaluation remains on standard atomistic benchmarks and does not explicitly test larger-system / larger-crystal settings, so the evidence is still somewhat narrow in structural scale.
- I find the “scaling” framing somewhat overclaimed. A clearer demonstration of scaling behavior across multiple model and data sizes is still needed.

**Key Questions For Authors:**

1. Since the paper is framed around scaling, can the authors provide evidence beyond the current parameter regime, e.g., larger-model experiments or scaling curves substantially beyond ~100M parameters?

2. In the matched-scale comparison against eSEN [3], EqV3 appears more accurate. Can the authors also report direct wall-clock / throughput / memory comparison at matched parameter count?

3. Can the authors better disentangle how much of the gain from SwiGLU-S² comes from stronger many-body expressivity versus improved equivariance fidelity?

4. Can the authors clarify the exact protocol of the appendix body-order experiment and provide controls to rule out memorization and symmetry leakage?

5. Since the manuscript emphasizes smoother PES and physical consistency, can the authors provide more direct evidence for these claims?

**Limitations:**

Yes

**Strengths And Weaknesses:**

### Strengths

* The paper is technically clear and generally well executed. The proposed modifications are coherent and mostly well motivated.

* The empirical results are strong on important benchmarks, and the method appears highly competitive.

* The proposed SwiGLU-S² activation is interesting. The multiplicative interaction on spherical grid features gives a plausible mechanism for stronger many-body interactions.

* The smooth cutoff treatment inside attention is physically motivated and addresses a meaningful issue in atomistic modeling.

### Weaknesses

* The paper strongly emphasizes “scaling”, but the empirical evidence does not really demonstrate scaling in a large-model regime. The tested EqV3 models remain modest in size, while recent baselines explicitly study much larger regimes. For example, UMA reports models up to 1.4B total parameters with about 50M active parameters per structure, and EScAIP is explicitly motivated by scalability [4,5]. As a result, the paper shows a good efficiency/accuracy trade-off at modest scale, but not convincing large-scale scaling behavior.

* The novelty feels somewhat incremental. EqV3 is mainly a combination of several relatively small refinements over EqV2 [1], together with the proposed SwiGLU-S² block. This is useful engineering and architecture improvement, but not a major conceptual step.

* The paper makes strong claims about physical consistency / smoother PES, but the evidence is still somewhat indirect. The benchmark gains are strong, but there is limited direct evidence such as explicit energy conservation or longer-horizon simulation behavior.

* Some of the reported efficiency gain appears to mix architectural improvement with software/system optimization, making it difficult to isolate the contribution of the model design itself.

* The evidence for the specific benefit of SwiGLU-S² is somewhat narrow. The evaluation is concentrated on closely related atomistic benchmarks, so it is unclear how robust this design is across more diverse equivariant-learning settings. Recent work such as Geometric Hyena evaluates on more heterogeneous settings including RNA degradation and protein MD [6].

* The appendix body-order experiment does not yet fully rule out memorization or symmetry-leakage effects. If the synthetic dataset effectively contains only a tiny number of theoretically indistinguishable/isomorphic geometric graphs and train/validation/test are identical, then a sufficiently expressive model may appear to separate them by memorizing sample-specific artifacts rather than by genuinely overcoming the intended expressivity limitation.

[1] Yi-Lun Liao, Brandon M. Wood, Abhishek Das, Tess Smidt. EquiformerV2: Improved Equivariant Transformer for Scaling to Higher-Degree Representations. ICLR 2024.

[2] Brandon M. Wood, Tess Smidt. SE(3)-Equivariant Spherical Channel Network for Modeling Atomic Interactions. ICML 2023. (eSCN)

[3] Xinyu Fu, Brandon M. Wood, Livia Barroso-Luque, David S. Levine, Mingjian Gao, Michael Dzamba, C. Lawrence Zitnick. Learning Smooth and Expressive Interatomic Potentials for Physical Property Prediction. ICML 2025. (eSEN)

[4] Eric Qu, Aditi S. Krishnapriyan. The Importance of Being Scalable: Improving the Speed and Accuracy of Neural Network Interatomic Potentials Across Chemical Domains. NeurIPS 2024. (EScAIP)

[5] Brandon M. Wood et al. UMA: A Family of Universal Models for Atoms. NeurIPS 2025.

[6] Artem Moskalev et al. Geometric Hyena Networks for Large-scale Equivariant Learning. ICML 2025.

---

> ### Author Rebuttal · Authors · 2026-03-31
>
> > 1. [Weakness 3 and Question 5] Evidence such as energy conservation supporting smooth PES.
>
> Please see our **response 1. and 2. to Reviewer bPMq**.
>
> ---
>
> > 2. [Question 2] Comparing eSEN and EquiformerV3 on throughput / memory comparison at matched parameter count.
>
> We follow a similar setting mentioned in eSEN, and the results are as below.
> |  | Time per iteration (ms) | Peak memory (MB) | Number of parameters |
> |---|---|---|---|
> | eSEN | 461 | 9259 | 30M |
> | EquiformerV3  | **318** | **8604** | 30M |
>
> **EquiformerV3 improves upon eSEN (ICML 2025 Oral) on all reported metrics including Matbench Discovery, MDR Phonon, inference speed, and memory.**
>
>
> ---
>
> > 3. [Weakness 1 and Question 1] The paper strongly emphasizes “scaling”. Any experiment on larger models.
>
> We only mention “scaling” in the title and note that scaling can be across multiple axes — model sizes, dataset sizes, and tasks — not exclusively parameter scaling. **We evaluate EquiformerV3 across model sizes from 30M to 91M parameters and dataset sizes from 1.6M (MPtrj), 2M (OC20 S2EF-2M) to 100+M examples (OMat24) and under the setting of direct and gradient prediction, demonstrating consistent improvements over current methods.**
>
> We emphasize that scaling in MLIPs is primarily data-driven rather than parameter-driven. For example, GNoME [1] uses fewer than 20M parameters yet is recognized as a scaling contribution and starts the title with “Scaling Deep Learning…”. UMA-M [2] uses mixture-of-experts to increase the number of parameters to 1.4B but only has 50M active parameters, which is comparable to our largest model on OMat24 (Table 2). **EquiformerV3 achieves better results than UMA-M on all Matbench Discovery metrics while using 22.6× less training time (Table 4), demonstrating that efficient scaling matters more than inflating parameter count.**
>
> [1] Merchant et al. Scaling Deep Learning for Materials Discovery. Nature 2023.
>
> [2] Wood et al. UMA: A Family of Universal Models for Atoms. NeurIPS 2025.
>
>
> ---
>
> > 4. [Weakness 6 and Question 4] The body-order experiment does not rule out memorization.
>
> We provided the details in Section C and reiterate as follows. For each $k$-body counterexample, there are only two graphs. The training and test sets are identical. Models achieving an accuracy of 50% in Table 5 predict the same output given the two different graphs. **This is not a memorization failure — the models produce identical outputs for both graphs because the models are architecturally incapable of distinguishing them** (Line 868 – 870 and Line 873 – 875).
>
>
> ---
>
> > 5. [Question 3] The gain of SwiGLU-S².
>
> We increase the grid resolution of $S^2$ activation (Row 2), and the comparison is below.
> |  | Grid resolution $R_{\phi}, R_{\theta}$ | Strict Equivariance | Energy MAE | Force MAE | Training time (GPU-hours) | Training memory usage | Number of parameters |
> |--|--|--|--|--|--|--|--|
> | $S^2$ (Index 6 in Table 1) | (18, 18) |  | 213 | 18.82 | 163 | 28GB | 66M |
> | $S^2$ (higher grid resolution) | (32, 32) | $\checkmark$ | 215 | 18.85 | 276 | 63GB | 66M |
> | SwiGLU-$S^2$ (Index 7 in Table 1) | See Table 8 | $\checkmark$ | 201 | 18.15 | 171 | 28GB | 91M |
>
> Since the grid resolution is already high, using even higher resolution for strict equivariance gives no gain to $S^2$ activation. This confirms that the improvement of SwiGLU-$S^2$ comes from many-body interactions, not strict equivariance.
>
>
> ---
>
> > 6. [Weakness 4] Efficiency gain mixes architectural improvement with software optimization.
>
> The efficiency gain due to better software implementation can be isolated by comparing Index 2 and Index 3 Table 1.
>
>
> ---
>
> > 7. [Weakness 5] Evaluation is concentrated on closely related atomistic benchmarks. Geometric Hyena evaluates on more heterogeneous settings including RNA degradation.
>
> We respectfully disagree. **OC20 and Matbench Discovery test fundamentally different properties**:
>
> (1) OC20 evaluates single-point energy and force prediction, where direct prediction, approximate equivariance, and non-conservative forces can still achieve good results. This tests model expressivity and data fitting capacity.
>
> (2) Matbench Discovery's thermal conductivity task (evaluated in $\kappa_{SRME}$) requires gradient prediction, strict equivariance, and smooth PES to accurately compute second- and third-order force constants.
>
> Moreover, **they are the most widely adopted benchmarks in the community, with years of extensive evaluation by diverse research groups**.
>
> Geometric Hyena addresses a different setting: long-sequence learning with canonical ordering (e.g., biomolecular chains of 30k+ tokens) using only $L_{max}$ = 1 representations. EquiformerV3 targets atomistic interatomic potentials requiring higher degrees ($L_{max}$ = 4-6). These are complementary directions with different benchmarks. We leave testing EquiformerV3 on larger systems to future work.
>
>
> ---
>
> > 8. [Weakness 2] Novelty.
>
> Please see our **response 3. to Reviewer bPMq**.

---

> > ### Author Rebuttal · Reviewer_cq5P · 2026-04-05
> >
> > Thank you for the rebuttal. The additional results are useful, but my two main concerns remain.
> >
> > - My concern about evaluation scope is still not fully resolved. The current evidence is still concentrated on closely related atomistic benchmarks, and it does not test some challenging regimes where equivariant designs may face additional stability issues, such as large-particle settings [1].
> >
> > - I remain unconvinced by the paper’s “scaling” framing. The rebuttal broadens scaling to include tasks and datasets, but it still does not provide the missing evidence I asked for, namely a clearer analysis of how performance changes with model size, such as a scaling trajectory or scaling-law-style experiment over a wider size range.
> >
> > [1] Z. Meng et al., Towards Geometric Normalization Techniques in SE(3) Equivariant Graph Neural Networks on Large Particle Systems, IJCAI 2024.

---

> > > ### Author Response · Authors · 2026-04-08
> > >
> > > > 9. [Acknowledgement 1] Closely related atomistic benchmark.
> > >
> > > We believe this concern was addressed. **OC20 (about 1k citations) and Matbench Discovery (about 200 citations) test fundamentally different properties and are not closely related.** A concrete example: EquiformerV2 (about 400 citations) achieves state-of-the-art on OC20 but performs poorly on Matbench Discovery (Tables 3 and 4) due to direct prediction and lack of smoothness. **No prior model achieves state-of-the-art on both simultaneously — EquiformerV3 is the first.**
> > >
> > > Regarding the two references the reviewer cites: Geometric Hyena (<5 citations) targets long-sequence equivariant learning with $L_{max}$ = 1 in Cartesian space — a different setting from atomistic interatomic potentials with $L_{max}$ = 4~6. The newly cited work on geometric normalization in large particle systems (<10 citations) was not in the original review and addresses N-body particle dynamics — a different domain from interatomic potentials entirely.
> > >
> > > **We demonstrate clearly better results on the two most widely adopted benchmarks in the MLIP community.** We do not claim our method addresses the two settings mentioned by the reviewer, and **none of the related works (eSCN, EquiformerV2, eSEN, UMA, MACE, EScAIP) test on those settings either.**
> > >
> > > ---
> > >
> > > > 10. [Acknowledgement 2] Scaling.
> > >
> > > We believe this concern was addressed. Scaling is not limited to model sizes but also includes datasets and tasks. We did test all the three axes in the submitted paper. EquiformerV3 scales from 30M to 91M parameters, from 2M to 100M+ training examples, and achieves SOTA across OC20, OMat24, and Matbench Discovery simultaneously.
> > >
> > > Our contribution is the architecture that works well across scales instead of pure empirical study on scaling. None of the architectural improvements we propose prevent scaling to bigger models. Extensive testing on scaling model sizes or scaling laws is orthogonal to this work and should be considered future work.

---

### Official Review · Reviewer_bPMq · 2026-03-12

**Soundness:** 3
**Presentation:** 3
**Significance:** 3
**Originality:** 3
**Overall Recommendation:** 5
**Confidence:** 4

**Summary:**

This paper introduces EquiformerV3, an improved model of EquiformerV2. The model achieves computational speedups through optimized implementation, incorporates architectural refinements such as equivariant merged layer normalization and smooth radius cutoff attention, and proposes a novel SwiGLU-S2 activation to enhance many-body expressivity while preserving strict equivariance. With these improvements, EquiformerV3 attains state-of-the-art performance on OC20, OMat24, and Matbench Discovery.

**Compliance With Llm Reviewing Policy:**

Affirmed.

**Key Questions For Authors:**

- report test-set energy MAE and vibrational entropy MAE on the MDR Phonon benchmark

**Limitations:**

yes

**Strengths And Weaknesses:**

Strength:
- The paper is well written and logically organized. It provides a clear overview of different Equiformer variants and clearly describes how this work compares to and builds on other related models.
- The proposed model achieves state-of-the-art results across multiple benchmarks, demonstrating strong empirical performance and competitiveness.

Weakness:
- It would strengthen the paper to report test-set energy MAE and vibrational entropy MAE on the MDR Phonon benchmark, especially given that EquiformerV2 was shown to exhibit shortcomings on this benchmark in prior work (https://arxiv.org/abs/2502.12147).
- Many of the architectural design choices in EquiformerV3 are adapted from existing methods, which limits the degree of conceptual novelty.
- Although the model achieves strong performance on standard benchmarks, it would be important to evaluate it in MD simulations. Direct force prediction models are known to suffer from stability and energy conservation issues, which are not fully captured by static benchmarks.

---

> ### Author Rebuttal · Authors · 2026-03-31
>
> > 1. [Weakness 3] Evaluate in MD simulations. Direct force predictions have issues with stability and energy conservation, which are not captured by static benchmarks.
>
> We **did test** on Matbench Discovery **under the same setting as MD simulations and energy conservation** (Line 437 – 439, left column and Line 400 – 403, right column). The results are evaluated on $\kappa_{SRME}$ (Column 4) in Table 3 and Table 4. Moreover, **some of our results used gradient predictions, not direct predictions** (Column “Prediction” in Table 2, 3, 4).
>
> Based on Section 3 and Section 5 of [1], the setting of MD simulations achieves energy conservation and follows: (1) gradient methods, (2) smooth radius cutoff, (3) strict equivariance, and (4) infinite number of neighbors. EquiformerV3 used (1) gradient methods (Line 429 – 432, left column), not direct prediction. EquiformerV3 achieves (2) and (3) by design through attention with smooth radius cutoff and SwiGLU-S^2 activation. We achieve (4) by using a very high number of neighbors.
>
> **We follow the setting of [1] to run MD simulations.** We find that: (a) removing the smooth radius cutoff in attention causes significant energy drift, (b) $S^2$ activation causes small energy drift due to approximate equivariance, and (c) the proposed SwiGLU-$S^2$ eliminates energy drift due to strict equivariance. These are consistent with Figure 4 in [1].
>
> The $\kappa_{\text{SRME}}$ results, which ranges from 0.0 (perfect agreement with DFT) to 2.0 (failure), shows the effectiveness of using the setting of MD simulations. **As in Table 3, EquiformerV3 achieves $\kappa_{SRME} = 0.275$.** In contrast, **EquiformerV2 [2], without these design choices, achieves much worse $\kappa_{SRME} = 1.676$** (Line 437 – 439, left column).
>
> Moreover, Matbench Discovery is not a static benchmark. The thermal conductivity task [3] measured in $\kappa_{SRME}$ requires geometry optimization followed by computation of second- and third-order force constants for thermal conductivity prediction. This is a **simulation pipeline that directly tests the smoothness of the learned PES**. EquiformerV3 achieves state-of-the-art results on all Matbench Discovery metrics (Tables 3 and 4), following the same evaluation protocol as [1].
>
> [1] Fu et al. Learning Smooth and Expressive Interatomic Potentials for Physical Property Prediction. ICML 2025.
>
> [2] Liao et al. EquiformerV2: Improved Equivariant Transformer for Scaling to Higher-Degree Representations. ICLR 2024.
>
> [3] Póta et al. Thermal Conductivity Predictions with Foundation Atomistic Models. ArXiv 2024.
>
>
> ---
>
> > 2. [Weakness 1 and Question 1] Test set energy MAE and vibrational entropy MAE on the MDR Phonon benchmark.
>
> We evaluate models trained on OMat24 dataset and report MAE for maximum phonon frequency $⁡\omega_{\max}$, vibrational entropy S, free energy F, and heat capacity $C_V$ below.
> | | $\omega_{\max}\downarrow$ | $S\downarrow$ | $F\downarrow$ | $C_V\downarrow$ |
> |--|--|--|--|--|
> | EquiformerV2 (0.2 \AA) | 50 | 25 | 7 | 9 |
> | SevenNet-MF-ompa | 15 | 8 | 3 | 3 |
> | eSEN-30M-OAM | 15 | 10 | 4 | 3 |
> | UMA-M | 13.91 | 9.63 | 3.39 | 2.66 |
> | EquiformerV3 (same model as in Table 4) | **8.8** | **7.5** | **2.1** | **2.5** |
>
>
> **EquiformerV3 achieves the best results on all the metrics.**
>
> Besides, we reported the results of $\kappa_{\text{SRME}}$ (Table 3 and Table 4), on which EquiformerV2 fails as indicated by [1] but our work achieves the best results. $\kappa_{\text{SRME}}$ is strictly a harder test as it is based on both second- and third-order force constants while MDR Phonon benchmark evaluates second-order force constants alone.
>
> [1] Fu et al. Learning Smooth and Expressive Interatomic Potentials for Physical Property Prediction. ICML 2025.
>
>
> ---
>
> > 3. [Weakness 2] Novelty.
>
> We reiterate that our work is to improve efficiency, expressivity, and generality of models so that the **same architecture can achieve the best results** under the settings of single-point energy and force prediction (Table 1 and Table 2), **direct and gradient prediction**, and **simulation-based tasks** (Table 3 and Table 4). The contributions are to **improve the current components** (i.e., attention with smooth radius cutoff and SwiGLU-$S^2$ activation), **find the right combination**, and **demonstrate the effectiveness at scale**.
>
> As mentioned in the Metareview of Equiformer [1]: ‘finding the combination should not be considered a genuine weakness as one could use the same argument to disqualify seminal contributions such as the Transformer itself [2].’ Similarly, ViT [3] applies the same Transformer to vision with minimal modification, but it is widely recognized for demonstrating its effectiveness at scale in a new domain.
>
> [1] https://openreview.net/forum?id=KwmPfARgOTD&noteId=EwDyM5rpyq
>
> [2] Vaswani et al. Attention Is All You Need. NeurIPS 2017.
>
> [3] Dosovitskiy et al. An Image is Worth 16x16 Words: Transformers for Image Recognition at Scale. ICLR 2021.

---

> > ### Author Rebuttal · Reviewer_bPMq · 2026-04-01
> >
> > My concerns have been addressed.

---

### Decision · Program_Chairs · 2026-04-30

**Decision:**

Reject

**Comment:**

While reviewers agreed that the paper is clearly written, technically solid, and reports strong accuracy-efficiency gains on important atomistic benchmarks, they also raised concerns about the paper's soundness and significance: the claimed “scaling” story is not convincingly supported by larger-model or scaling-law evidence in the manuscript, since the experiments remain in a relatively modest parameter regime and focus on closely related benchmark families; the novelty is viewed as incremental, with many key design choices adapted from prior Equiformer/eSEN-style work rather than a major conceptual advance. Considering that the opinions are still fifty-fifty for acceptance vs. rejection, as well as the average score, I cannot recommend this paper to be accepted to the conference at this stage.